# COMPARING REWINDING AND FINE-TUNING IN NEURAL NETWORK PRUNING

**Alex Renda**
MIT CSAIL
renda@csail.mit.edu

**Jonathan Frankle**
MIT CSAIL
jfrankle@csail.mit.edu

**Michael Carbin**
MIT CSAIL
mcarbin@csail.mit.edu

## ABSTRACT

Many neural network pruning algorithms proceed in three steps: train the network to completion, remove unwanted structure to compress the network, and retrain the remaining structure to recover lost accuracy. The standard retraining technique, *fine-tuning*, trains the unpruned weights from their final trained values using a small fixed learning rate. In this paper, we compare fine-tuning to alternative retraining techniques. *Weight rewinding* (as proposed by Frankle et al. (2019)), rewinds unpruned weights to their values from earlier in training and retrains them from there using the original training schedule. *Learning rate rewinding* (which we propose) trains the unpruned weights from their final values using the same learning rate schedule as weight rewinding. Both rewinding techniques outperform fine-tuning, forming the basis of a network-agnostic pruning algorithm that matches the accuracy and compression ratios of several more network-specific state-of-the-art techniques.

## 1 INTRODUCTION

Pruning is a set of techniques for removing weights, filters, neurons, or other structures from neural networks (e.g., Le Cun et al., 1990; Reed, 1993; Han et al., 2015; Li et al., 2017; Liu et al., 2019). Pruning can compress standard networks across a variety of tasks, including computer vision and natural language processing, while maintaining the accuracy of the original network. Doing so can reduce the parameter count and resource demands of neural network inference by decreasing storage requirements, energy consumption, and latency (Han, 2017).

We identify two classes of pruning techniques in the literature. One class, exemplified by *regularization* (Louizos et al., 2018) and *gradual pruning* (Zhu & Gupta, 2018; Gale et al., 2019), prunes the network throughout the standard training process, producing a pruned network by the end of training.

The other class, exemplified by *retraining* (Han et al., 2015), prunes after the standard training process. Specifically, when parts of the network are removed during the pruning step, accuracy typically decreases (Han et al., 2015). It is therefore standard to *retrain* the pruned network to recover accuracy. Pruning and retraining can be repeated *iteratively* until a target sparsity or accuracy threshold is met; doing so often results in higher accuracy than pruning in *one shot* (Han et al., 2015). A single iteration of the retraining based pruning algorithm proceeds as follows (Liu et al., 2019):

1. TRAIN the network to completion.
2. PRUNE structures of the network, chosen according to some heuristic.
3. RETRAIN the network for some time ($t$ epochs) to recover the accuracy lost from pruning.

The most common retraining technique, *fine-tuning*, trains the pruned weights for a further $t$ epochs at a fixed learning rate (Han et al., 2015), often the final learning rate from training (Liu et al., 2019).

Work on the *lottery ticket hypothesis* introduces a new retraining technique, *weight rewinding* (Frankle et al., 2019), although Frankle et al. do not evaluate it as such. The lottery ticket hypothesis proposes that early in training, sparse subnetworks emerge that can train in isolation to the same accuracy as the original network (Frankle & Carbin, 2019). To find such subnetworks, Frankle et al. (2019) propose training to completion and pruning (steps 1 and 2 above) and then *rewinding* the unpruned

weights by setting their values back to what they were earlier in training. If this pruned and rewound subnetwork trains to the same accuracy as the original network (reusing the original learning rate schedule), then—for their purposes—this validates that such trainable subnetworks exist early in training. For our purposes, this rewinding and retraining technique is simply another approach for retraining after pruning. The selection of where to rewind the weights to is controlled by the retraining time $t$; retraining for $t$ epochs entails rewinding to $t$ epochs before the end of training.

We also propose a new variation of weight rewinding, *learning rate rewinding*. While weight rewinding rewinds both the weights and the learning rate, learning rate rewinding rewinds only the learning rate, continuing to train the weights from their values at the end of training (like fine-tuning). This is similar to the learning rate schedule used by cyclical learning rates (Smith, 2017).

In this paper, we compare fine-tuning, weight rewinding, and learning rate rewinding as retraining techniques after pruning. We evaluate these techniques according to three criteria:

ACCURACY      The accuracy of the resulting pruned network.
EFFICIENCY    The resources required to represent or execute the pruned network.
SEARCH COST   The cost to find the pruned network (i.e., the amount of retraining required).

The goal of neural network pruning is to increase EFFICIENCY while maintaining ACCURACY. In this paper we specifically study PARAMETER-EFFICIENCY, the parameter count of the pruned neural network.[1] We also evaluate the SEARCH COST of finding the pruned network, measured as the number of epochs for which the network is retrained.

We compare the pruning and retraining techniques evaluated in this paper against pruning algorithms from the literature that are shown to be state-of-the-art by Ortiz et al. (2020). These state-of-the-art algorithms are complex to use, requiring network-specific hyperparameters (Carreira-Perpiñán & Idelbayev, 2018; Zhu & Gupta, 2018) or reinforcement learning (He et al., 2018).

**Contributions.**

- We show that retraining with weight rewinding outperforms retraining with fine-tuning across networks and datasets. When rewinding to anywhere within a wide range of points throughout training, weight rewinding is a drop-in replacement for fine-tuning that achieves higher ACCURACY for equivalent SEARCH COST.

- We propose a simplification of weight rewinding, *learning rate rewinding*, which rewinds the learning rate schedule but not the weights. Learning rate rewinding matches or outperforms weight rewinding in all scenarios.

- We propose a pruning algorithm based on learning rate rewinding with network-agnostic hyperparameters that matches state-of-the-art tradeoffs between ACCURACY and PARAMETER-EFFICIENCY across networks and datasets. The algorithm proceeds as follows: 1) train to completion, 2) globally prune the $20\%$ of weights with the lowest magnitudes, 3) retrain with learning rate rewinding for the full original training time, and 4) iteratively repeat steps 2 and 3 until the desired sparsity is reached.

- We find that weight rewinding can nearly match the ACCURACY of this proposed pruning algorithm, meaning that lottery tickets found by pruning and rewinding are state-of-the-art pruned networks.

We show that learning rate rewinding outperforms the standard practice of fine-tuning without requiring any network-specific hyperparameters in all settings that we study. This technique forms the basis of a simple, state-of-the-art pruning algorithm that we propose as a valuable baseline for future research and as a compelling default choice for pruning in practice.

## 2    METHODOLOGY

In this paper, we evaluate weight rewinding and learning rate rewinding as retraining techniques. We therefore do not consider regularization or gradual pruning techniques, except when comparing against state-of-the-art. Creating a retraining based pruning algorithm involves instantiating each

---

[1]We discuss other instantiations of EFFICIENCY, such as FLOPs, in Section 6 and Appendix F.

of the steps in Section 1 (TRAIN, PRUNE, RETRAIN) from a range of choices. Below, we discuss the set of design choices considered in our experiments and mention other standard choices. Our implementation and the data from the experiments in this paper are available at: https://github.com/lottery-ticket/rewinding-iclr20-public

## 2.1 HOW DO WE TRAIN?

We assume that TRAIN is provided as the standard training schedule for a network. Here, we discuss the networks, datasets, and training hyperparameters used in the experiments in this paper.

We study neural network pruning on a variety of standard architectures for image classification and machine translation. Specifically, we consider ResNet-56 (He et al., 2016) for CIFAR-10 (Krizhevsky, 2009), ResNet-34 and ResNet-50 (He et al., 2016) for ImageNet (Russakovsky et al., 2015), and GNMT (Wu et al., 2016) for WMT16 EN-DE. Our implementations and hyperparameters are from standard reference implementations, as described in Table 1, with the exception of the GNMT model. For GNMT, we extend the training schedule used in the reference implementation to reach standard BLEU scores on the validation set, rather than the lower BLEU reached by the reference implementation.[2] This extended schedule uses the same standard GNMT warmup and decay schedule as the original training schedule (Luong et al., 2017), but expanded to span 5 epochs rather than 2.

| Dataset | Network | #Params | Optimizer | Learning rate (t = training epoch) | Test accuracy |
|---------|---------|---------|-----------|-----------------------------------|---------------|
| CIFAR-10 | ResNet-56[3] | 852K | Nesterov SGD $\beta = 0.9$ Batch size: 128 Weight decay: 0.0002 Epochs: 182 | $\alpha = \begin{cases} 0.1 & t \in [0, 91) \\ 0.01 & t \in [91, 136) \\ 0.001 & t \in [136, 182] \end{cases}$ | $93.46 \pm 0.21\%$ |
| ImageNet | ResNet-34[4] | 21.8M | Nesterov SGD $\beta = 0.9$ Batch size: 1024 Weight decay: 0.0001 Epochs: 90 | $\alpha = \begin{cases} 0.4 \cdot \frac{t}{5} & t \in [0, 5) \\ 0.4 & t \in [5, 30) \\ 0.04 & t \in [30, 60) \\ 0.004 & t \in [60, 80) \\ 0.0004 & t \in [80, 90] \end{cases}$ | $73.60 \pm 0.27\%$ top-1 |
| | ResNet-50[4] | 25.5M | | | $76.17 \pm 0.14\%$ top-1 |
| WMT16 EN-DE | GNMT[5] | 165M | Adam $\beta_1 = 0.9$ $\beta_2 = 0.999$ Batch size: 2048 Epochs: 5 | $\alpha = \begin{cases} 0.002 \cdot 0.01^{1-8t} & t \in [0, 0.125) \\ 0.002 & t \in [0.125, 3.75) \\ 0.001 & t \in [3.75, 4.165) \\ 0.0005 & t \in [4.165, 4.58) \\ 0.00025 & t \in [4.58, 5) \end{cases}$ | newstest2015: $26.87 \pm 0.23$ BLEU |

Table 1: Networks, datasets, and hyperparameters. We use standard implementations available online and standard hyperparameters. All accuracies are in line with baselines reported for these networks (Liu et al., 2019; He et al., 2018; Gale et al., 2019; Wu et al., 2016; Zhu & Gupta, 2018).

## 2.2 HOW DO WE PRUNE?

**What structure do we prune?**
UNSTRUCTURED PRUNING. Unstructured pruning prunes individual weights without consideration for where they occur within each tensor (e.g., Le Cun et al., 1990; Han et al., 2015).

STRUCTURED PRUNING. Structured pruning involves pruning weights in groups, removing neurons, convolutional filters, or channels (e.g., Li et al., 2017).

Unstructured pruning reduces the number of parameters, but may not improve performance on commodity hardware until a large fraction of weights have been pruned (Park et al., 2017). Structured pruning preserves dense computation, meaning that it can lead to immediate performance improvements (Liu et al., 2017). In this paper, we study both unstructured and structured pruning.

**What heuristic do we use to prune?**
MAGNITUDE PRUNING. Pruning weights with the lowest magnitudes (Han et al., 2015) is a standard

---

[2]The reference implementation is from MLPerf 0.5 (Mattson et al., 2020) and reaches a newstest-2014 BLEU of 21.8. With the extended training schedule, we reach a more standard BLEU of 24.2 (Wu et al., 2016).

[3]https://github.com/tensorflow/models/tree/v1.13.0/official/resnet

[4]https://github.com/tensorflow/tpu/tree/98497e0b/models/official/resnet

[5]https://github.com/mlperf/training_results_v0.5/tree/7238ee7/v0.5.0/google/cloud_v3.8/gnmt-tpuv3-8

choice that achieves state-of-the-art ACCURACY versus EFFICIENCY tradeoffs (Gale et al., 2019). For unstructured pruning, we prune the lowest magnitude weights *globally* throughout the network (Lee et al., 2019; Frankle & Carbin, 2019). For structured pruning, we prune convolutional filters by their $L_1$ norms using the per-layer pruning rates hand-chosen by Li et al. (2017).[6] Specifically, we study ResNet-56-B and ResNet-34-A from Li et al. (2017).

We only consider magnitude-based pruning heuristics in this paper, although there are a wide variety of other pruning heuristics in the literature, including those that learn which weights to prune as part of the optimization process (e.g., Louizos et al., 2018; Molchanov et al., 2017) and those that prune based on other information (e.g., Le Cun et al., 1990; Theis et al., 2018; Lee et al., 2019).

### 2.3    HOW DO WE RETRAIN?

Let $W_g \in \mathbb{R}^d$ be the weights of the network at epoch $g$. Let $m \in \{0, 1\}^d$ be the pruning mask, such that the element-wise product $W \odot m$ denotes the pruned network. Let $T$ be the number of epochs that the network is trained for. Let $S[g]$ be the learning rate for each epoch $g$, defined such that $S[g > T] = S[T]$ (i.e., the last learning rate is extended indefinitely). Let $\text{TRAIN}^t(W, m, g)$ be a function that trains the network $W \odot m$ for $t$ epochs according to the original learning rate schedule $S$, starting from epoch $g$.

FINE-TUNING. Fine-tuning retrains the unpruned weights from their final values for a specified number of epochs $t$ using a fixed learning rate. Fine-tuning is the current standard practice in the literature (Han et al., 2015; Liu et al., 2019). It is typical to fine-tune using the last learning rate of the original training schedule (Li et al., 2017; Liu et al., 2019), a convention we follow in our experiments. Other choices are possible, including those found through hyperparameter search (Han et al., 2015; Han, 2017; Guan et al., 2019). Formally, fine-tuning for $t$ epochs runs $\text{TRAIN}^t(W_T, m, T)$.

WEIGHT REWINDING. Weight rewinding retrains by *rewinding* the unpruned weights to their values from $t$ epochs earlier in training and subsequently retraining the unpruned weights from there. It also rewinds the learning rate schedule to its state from $t$ epochs earlier in training. Retraining with weight rewinding therefore depends on the hyperparameter choices made during the initial training phase of the unpruned network. Weight rewinding was proposed to study the lottery ticket hypothesis by Frankle et al. (2019). Formally, weight rewinding for $t$ epochs runs $\text{TRAIN}^t(W_{T-t}, m, T - t)$.

LEARNING RATE REWINDING. Learning rate rewinding is a hybrid between fine-tuning and weight rewinding. Like fine-tuning, it uses the final weight values from the end of training. However, when retraining for $t$ epochs, learning rate rewinding uses the learning rate schedule from the last $t$ epochs of training (what weight rewinding would use) rather than the final learning rate from training (what fine-tuning would use). Formally, learning rate rewinding for $t$ epochs runs $\text{TRAIN}^t(W_T, m, T - t)$. We propose learning rate rewinding in this paper as a novel retraining technique.

In this paper, we compare all three retraining techniques. For each network, we consider ten retraining times $t$ evenly distributed between 0 epochs and the number of epochs for which the network was originally trained. For iterative pruning, this retraining time is ran per pruning iteration.

### 2.4    DO WE PRUNE ITERATIVELY?

ONE-SHOT PRUNING. The outline above prunes the network to a target sparsity level all at once, known as one-shot pruning (Li et al., 2017; Liu et al., 2019).

ITERATIVE PRUNING. An alternative is to iterate steps 2 and 3, pruning weights (step 2), retraining (step 3), pruning more weights, retraining further, etc., until a target sparsity level is reached. Doing so is known as iterative pruning. In practice, iterative pruning typically makes it possible to prune more weights while maintaining accuracy (Han et al., 2015; Frankle & Carbin, 2019).

In this paper, we consider both one-shot and iterative pruning. When running iterative pruning, we prune 20% of weights per iteration (Frankle & Carbin, 2019). When iteratively pruning with weight rewinding, weights are always rewound to the same values $W_{T-t}$ from the original run of training.

---

[6]To study multiple sparsity levels using these hand-chosen rates, we extrapolate these per-layer pruning rates to higher levels of sparsity, by exponentiating each per-layer pruning rate $p_i$ (which denotes the resulting density of layer $i$) by $k \in \{1, 2, 3, 4, 5\}$, creating new per-layer pruning rates $p_i^k$.

## 2.5 METRICS

We evaluate a pruned network according to three criteria.

ACCURACY is the performance of the pruned network on unseen data from the same distribution as the training set (i.e., the validation or test set). Higher accuracy values indicate better performance, and a typical goal is to match the accuracy of the unpruned network. All plots show the median, minimum, and maximum test accuracies reached across three different training runs.

For vision networks, we use $20\%$ of the original test set, selected at random, as the validation set; the remainder of the original test set is used to report test accuracies. For WMT16 EN-DE, we use `newstest2014` as the validation set (following Wu et al., 2016), and `newstest2015` as the test set (following Zhu & Gupta, 2018).

EFFICIENCY is the resources required to represent or perform inference with the pruned network. This can take multiple forms. We study PARAMETER-EFFICIENCY, the parameter count of the network. We specifically measure PARAMETER-EFFICIENCY relative to the full network with the *compression ratio* of the pruned network. For instance, if the pruned network has $5\%$ of weights remaining, then its compression ratio is $20\times$. Higher compression ratios indicate better PARAMETER-EFFICIENCY. We discuss other instantiations of EFFICIENCY in Section 6 and Appendix F

SEARCH COST is the computational resources required to find the pruning mask and retrain the remaining weights. We approximate SEARCH COST using *retraining time*, the total number of additional retraining epochs. Fewer retraining epochs indicates a lower SEARCH COST. Note that this metric does not consider speedup from retraining pruned networks. For instance, a network pruned to $20\times$ compression may be faster to retrain than if only pruned to $2\times$ compression.

## 3 ACCURACY VERSUS PARAMETER-EFFICIENCY TRADEOFF

In this section, we consider the Pareto frontier of the tradeoff between ACCURACY and PARAMETER-EFFICIENCY using each retraining technique, without regard for SEARCH COST. In other words, we study the highest accuracy each retraining technique can achieve at each compression ratio. We find that weight rewinding can achieve higher accuracy than fine-tuning across compression ratios on all studied networks and datasets. We further find that learning rate rewinding matches or outperforms weight rewinding in all scenarios. With iterative unstructured pruning, learning rate rewinding achieves state-of-the-art ACCURACY versus PARAMETER-EFFICIENCY tradeoffs, and weight rewinding remains close.

**Methodology.** For each retraining technique, network, and compression ratio, we select the setting of retraining time with the highest validation accuracy and plot the corresponding test accuracy.

**One-shot pruning results.** Figure 1 presents the results for one-shot pruning. At low compression ratios (when all techniques match the accuracy of the unpruned network), there is little differentiation between the techniques. However, learning rate rewinding typically results in higher accuracy than the unpruned network, whereas other techniques only match the original accuracy. At higher compression ratios (when no techniques match the unpruned network accuracy), there is more differentiation between the techniques, with fine-tuning losing more accuracy than either rewinding technique. Weight rewinding outperforms fine-tuning in all scenarios. Learning rate rewinding in turn outperforms weight rewinding by a small margin.

**Iterative pruning results.** Figure 2 presents the results for iterative unstructured pruning. As a basis for comparison, we also plot the drop in accuracy achieved by state-of-the-art techniques (as described in Appendix C and shown to be state-of-the-art by Ortiz et al. (2020)) as individual black dots. In iterative pruning, weight rewinding continues to outperform fine-tuning, and learning rate rewinding continues to outperform weight rewinding. Learning rate rewinding matches the ACCURACY versus PARAMETER-EFFICIENCY tradeoffs of state-of-the-art techniques across all datasets. In particular, learning rate rewinding with iterative unstructured pruning produces a ResNet-50 that matches the accuracy of the original network at $5.96\times$ compression, to the best of our knowledge a new state-of-the-art ResNet-50 compression ratio with no drop in accuracy. Weight rewinding nearly matches these state-of-the-art results, with the exception of high compression ratios on GNMT.

**One-shot ACCURACY versus PARAMETER-EFFICIENCY Tradeoff**

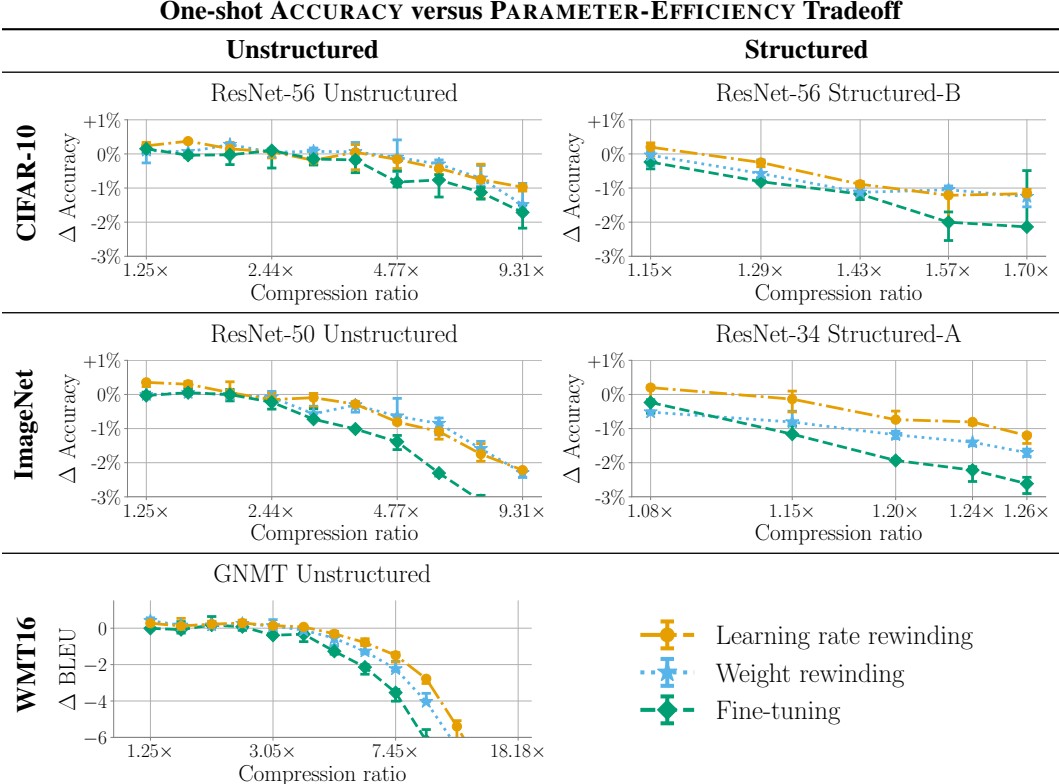

Figure 1: The best achievable accuracy across retraining times by one-shot pruning.

**Iterative ACCURACY versus PARAMETER-EFFICIENCY Tradeoff**

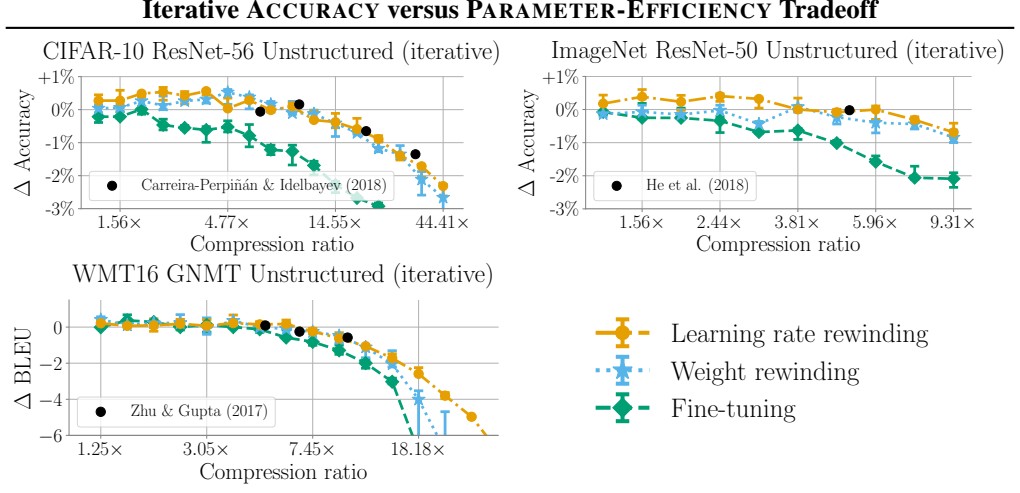

Figure 2: The best achievable accuracy across retraining times by iteratively pruning.

**Takeaway.** Retraining with weight rewinding outperforms retraining with fine-tuning across networks and datasets. Learning rate rewinding in turn matches or outperforms weight rewinding in all scenarios. Combined with iterative unstructured pruning, learning rate rewinding matches the tradeoffs between ACCURACY and PARAMETER-EFFICIENCY achieved by more complex techniques. Weight rewinding nearly matches these state-of-the-art tradeoffs.

## 4 ACCURACY VERSUS SEARCH COST TRADEOFF

In this section, we consider the tradeoff between ACCURACY and SEARCH COST for each retraining technique across a selection of compression ratios. In other words, we consider each method's ACCURACY given a fixed SEARCH COST. We show that both rewinding techniques achieve higher accuracy than fine-tuning for a variety of different retraining times $t$ (corresponding to different SEARCH COSTS). Therefore, in many contexts either rewinding technique can serve as a drop-in replacement for fine-tuning and achieve higher accuracy. Moreover, we find that using learning rate rewinding and retraining for the full training time of the original network leads to the highest accuracy among all tested retraining techniques, simplifying the hyperparameter search process.

**Methodology.** Figures 3 (unstructured pruning) and 4 (structured pruning) show the accuracy of each retraining technique as we vary the amount of retraining time; that is, the tradeoff between ACCURACY and SEARCH COST. Each plot shows this tradeoff at a specific compression ratio. The left column shows comparisons for *No Accuracy Drop*, which we define as the highest compression ratio at which any retraining technique can match the accuracy of the original network for any amount of SEARCH COST. The right column shows comparisons for *1%/1 BLEU Accuracy Drop*, which we define as the highest compression ratio at which any retraining technique gets within $1\%$ accuracy or 1 BLEU of the original network. We include similar plots for all tested compression ratios in Appendix E. All results presented in this section are for one-shot pruning; Appendix E also includes iterative pruning results, which exhibit the same trends.

**Unstructured pruning results.** Both rewinding techniques almost always match or outperform fine-tuning for equivalent retraining epochs. The sole exception is using weight rewinding and retraining for the full original training time, thereby rewinding the weights to the beginning of training: Frankle et al. (2019) show that accuracy drops if weights are rewound too close to initialization, and we find the same behavior here. We define the *rewinding safe zone* as the maximal region (as a percentage of original training time) across all networks in which both forms of rewinding outperform fine-tuning for an equivalent SEARCH COST. This zone (shaded gray in Figure 3) occurs when retraining for $25\%$ to $90\%$ of the original training time. Within this region, either rewinding technique can serve as a drop-in replacement for fine-tuning.

With learning rate rewinding, retraining for longer almost always results in higher accuracy. The same is true for weight rewinding other than when weights are rewound to near the beginning of training. On most networks and compression ratios, accuracy from rewinding saturates after retraining for roughly half of the original training time: while accuracy can continue to increase with more retraining, this gain is limited.

**Structured pruning results.** Structured pruning exhibits the same trends as unstructured pruning,[7] except that retraining with weight rewinding does not result in a drop in accuracy when retraining for the full training time (thereby rewinding to the beginning of training). This is consistent with the findings of Liu et al. (2019), who show that fine-tuning after structured pruning provides no accuracy advantage over reinitializing and training the pruned network from scratch. Liu et al. (2019) indicate that initialization is less consequential for retraining after structured pruning than for it is for retraining after unstructured pruning. Since weight rewinding and learning rate rewinding only differ in initialization before retraining, we expect and observe that they achieve similar accuracies when used for retraining after structured pruning.

**Takeaway.** Both weight rewinding and learning rate rewinding outperform fine-tuning across a wide range of retraining times, thereby serving as drop-in replacements that achieve higher accuracy anywhere within the rewinding safe zone. To achieve the most accurate network, retrain with learning rate rewinding for the full original training time (although accuracy saturates after retraining for about half of the original training time).

---

[7]On the CIFAR-10 ResNet-56 Structured-B at $1\%$ Accuracy Drop, we find that learning rate rewinding reaches lower accuracy than fine-tuning when retraining for 30 epochs. At this retraining time, the techniques are identical: the learning rate in the last 30 epochs of training, which learning rate rewinding uses, is the same as the final learning rate, which fine-tuning uses. The observed accuracy difference at that point therefore appears to be a result of random noise, and is not characteristic of the retraining techniques.

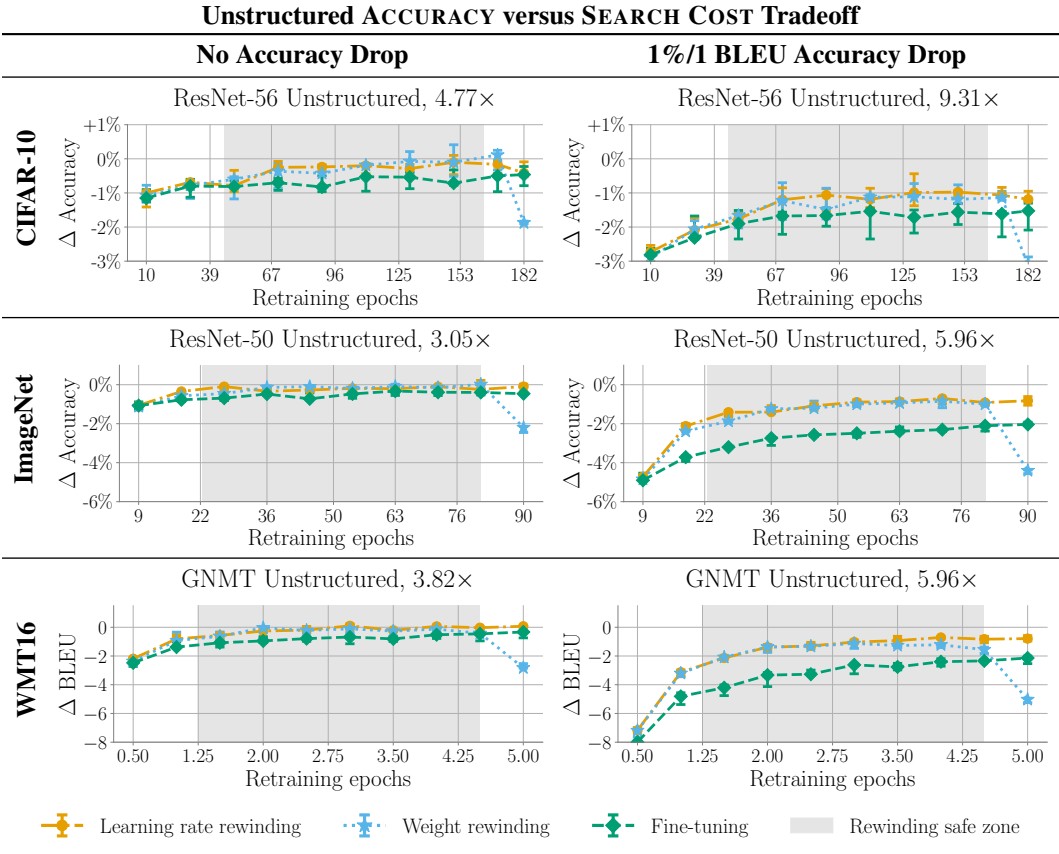

Figure 3: Accuracy curves across networks and compression ratios using unstructured pruning.

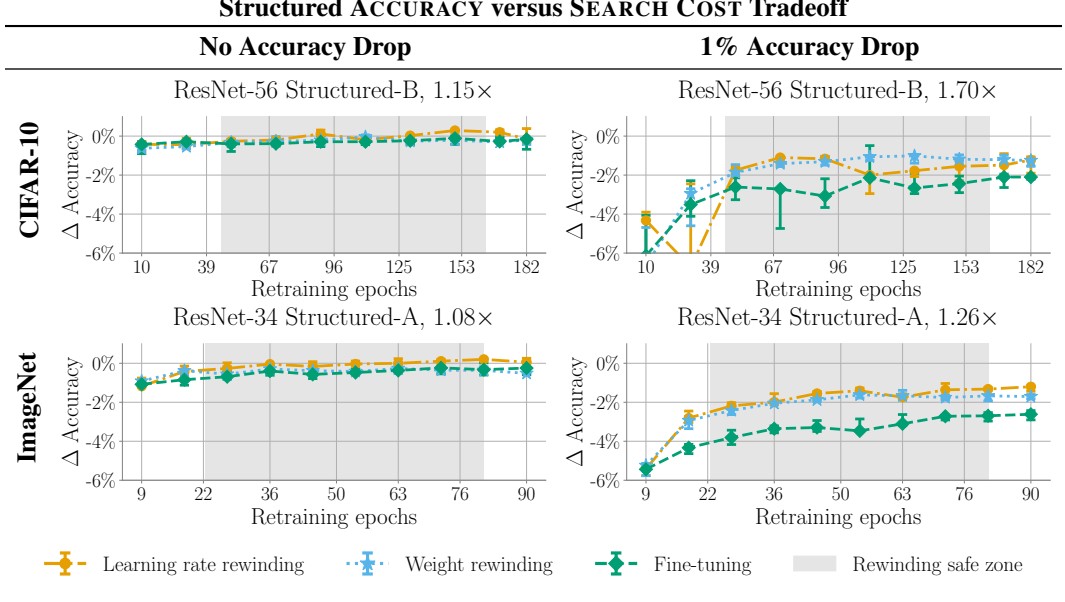

Figure 4: Accuracy curves across networks and compression ratios using structured pruning.

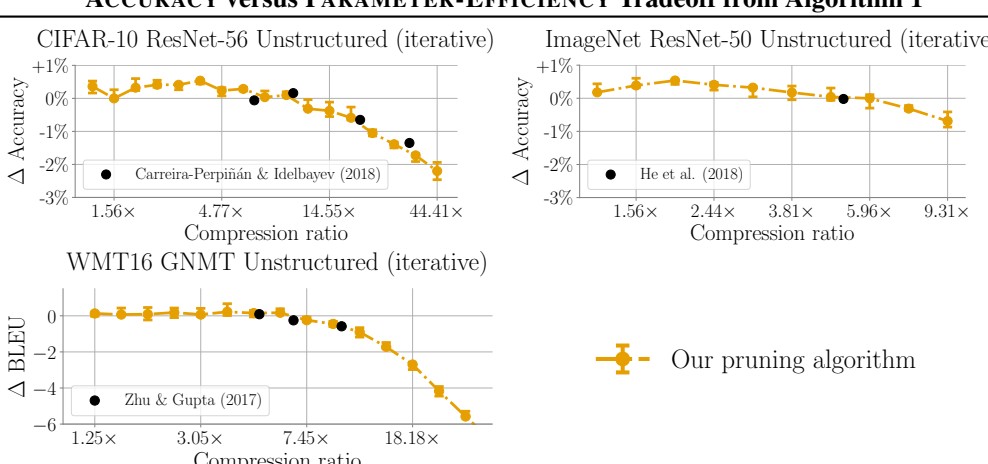

Figure 5: ACCURACY versus PARAMETER-EFFICIENCY tradeoff of our pruning algorithm.

## 5  OUR PRUNING ALGORITHM

Based on the results in Sections 3 and 4, we propose a pruning algorithm that is on the state-of-the-art ACCURACY versus PARAMETER-EFFICIENCY Pareto frontier. Algorithm 1 presents an instantiation of the pruning algorithm from Section 2 using network-agnostic hyperparameters:

---

**Algorithm 1** Our pruning algorithm

1. TRAIN to completion.
2. PRUNE the 20% lowest-magnitude weights globally.
3. RETRAIN using learning rate rewinding for the original training time.
4. Repeat steps 2 and 3 iteratively until the desired compression ratio is reached.

---

Figure 5 presents an evaluation of our pruning algorithm. Specifically, we compare the ACCURACY versus PARAMETER-EFFICIENCY tradeoff achieved by our pruning algorithm and by state-of-the-art baselines. This results in the same state-of-the-art behavior seen in Section 3, without requiring any per-compression-ratio hyperparameter search. We evaluate other retraining methods in Appendix D.

The hyperparameters for our pruning algorithm are shared across all networks and tasks we consider: there are neither layer-wise pruning rates nor a pruning schedule to select, beyond the network-agnostic 20% per-iteration pruning rate from prior work (Frankle & Carbin, 2019). Moreover, our pruning algorithm matches the accuracy of pruning algorithms that require more hyperparameters and/or additional methods, such as reinforcement learning (He et al., 2018; Carreira-Perpiñán & Idelbayev, 2018; Zhu & Gupta, 2018).

## 6  DISCUSSION

**Weight rewinding.** When retraining with weight rewinding, the weights are rewound to their values from early in training. This means that after retraining with weight rewinding, the weights themselves receive no more gradient updates than in the original training phase. Nevertheless, weight rewinding outperforms fine-tuning and is competitive with learning rate rewinding, losing little accuracy even though it reverts most of training. These results show that when pruning, it is not necessary to train the weights for a large number of steps; the pruning mask itself is a valuable output of pruning.

**Learning rate rewinding.** We propose learning rate rewinding, an alternative retraining technique that achieves state-of-the-art ACCURACY versus PARAMETER-EFFICIENCY tradeoffs. In this paper we do not investigate why the learning rate schedule used by learning rate rewinding achieves higher

accuracy than that of the standard fine-tuning schedule. We hope that further work on the optimization of sparse neural networks can shed light on why learning rate rewinding achieves higher accuracy than standard fine-tuning and can help derive other techniques for the training of sparse networks (Smith, 2017; Dettmers & Zettlemoyer, 2019).

The retraining techniques we consider reuse the hyperparameters from the original training process. This choice inherently narrows the design space of retraining techniques by coupling the learning rate schedule of retraining to that of the original training process. There may be further opportunities to improve performance by decoupling the hyperparameters of training and retraining and considering other retraining learning rate schedules. However, these potential opportunities come with the cost of added hyperparameter search.

**SEARCH COST.** Achieving state-of-the-art ACCURACY versus PARAMETER-EFFICIENCY tradeoffs with our pruning algorithm requires substantial SEARCH COST. Our pruning algorithm requires $T \cdot (1 + k)$ total training epochs to reach compression ratio $1 / 0.8^k$, where $T$ is the original network training time, and $k$ is the number of pruning iterations. In contrast, on CIFAR-10 ($T = 182$ epochs) Carreira-Perpiñán & Idelbayev (2018) employ a gradual pruning technique followed by fine-tuning, training for a total of 317 epochs to reach any compression ratio. On ImageNet ($T = 90$ epochs), He et al. (2018) retrain the ResNet-50 for 376 epochs to match the accuracy of the original network at $5.13\times$ compression. On WMT-16 ($T = 5$ epochs), Zhu & Gupta (2018) use a gradual pruning technique that trains and prunes over the course of about 11 epochs to reach any compression ratio.

The SEARCH COSTS of these other methods do not take into account the per-network hyperparameter search that each method required to find the settings that produced the reported results, nor the cost of the pruning heuristics themselves (e.g., training a reinforcement learning agent to predict pruning rates). In addition to optimizing ACCURACY and PARAMETER-EFFICIENCY, we believe that pruning research should also consider SEARCH COST (including hyperparameter search and training time).

**EFFICIENCY.** In this paper we study PARAMETER-EFFICIENCY: the number of parameters in the network. This provides a notion of scale of the network (Rosenfeld et al., 2020) and can serve as an input for theoretical analyses (Arora et al., 2018). There are other useful forms of EFFICIENCY that we do not study in this paper. One commonly studied form is INFERENCE-EFFICIENCY, the cost of performing inference with the pruned network. This is often measured in floating point operations (FLOPs) or wall clock time (Han, 2017; Han et al., 2016a). In Sections 3 and 4, we demonstrate that both rewinding techniques outperform fine-tuning after structured pruning (which explicitly targets INFERENCE-EFFICIENCY). In Appendix F, we show that iterative unstructured pruning and retraining with either rewinding technique results in networks that require fewer FLOPs to execute than those found by iterative unstructured pruning and retraining with fine-tuning.

Other forms of EFFICIENCY include STORAGE-EFFICIENCY (Han et al., 2016b), COMMUNICATION-EFFICIENCY (Alistarh et al., 2017), and ENERGY-EFFICIENCY (Yang et al., 2017).

**The Lottery Ticket Hypothesis.** Weight rewinding was first proposed by work on the *lottery ticket hypothesis* (Frankle & Carbin, 2019; Frankle et al., 2019), which studies the existence of sparse subnetworks that can train in isolation to full accuracy from near initialization. We present the first detailed comparison between the performance of these lottery ticket networks and pruned networks generated by standard fine-tuning. From this perspective, our results show that the sparse, lottery ticket networks that Frankle et al. (2019) uncover from early in training using weight rewinding can train to full accuracy at compression ratios that are competitive for pruned networks in general.

## 7 CONCLUSION

We find that both weight rewinding and learning rate rewinding outperform fine-tuning as techniques for retraining after pruning. When we perform iterative unstructured pruning and retrain with learning rate rewinding for the full original training time, we match the ACCURACY versus PARAMETER-EFFICIENCY tradeoffs of more complex techniques requiring network-specific hyperparameters. We believe that this algorithm is a valuable baseline for future research and a compelling default choice for pruning in practice.

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

## A  ACKNOWLEDGEMENTS

We gratefully acknowledge the support of Google, which provided us with access to the TPU resources necessary to conduct experiments on ImageNet and WMT through the TensorFlow Research Cloud. In particular, we express our gratitude to Zak Stone.

We gratefully acknowledge the support of IBM, which provided us with access to the GPU resources necessary to conduct experiments on CIFAR-10 through the MIT-IBM Watson AI Lab. In particular, we express our gratitude to David Cox and John Cohn.

This work was supported in part by cloud credits from the MIT Quest for Intelligence.

This work was supported in part by the Office of Naval Research (ONR N00014-17-1-2699).

This work was supported in part by DARPA Award #HR001118C0059.

## B  APPENDIX TABLE OF CONTENTS

## C  STATE-OF-THE-ART BASELINES

In Sections 3 and 5, we compare rewinding against state-of-the-art techniques from the literature. In this section, we describe the techniques we compare against. We specifically consider the techniques from the literature on the Pareto frontier of the ACCURACY versus PARAMETER-EFFICIENCY curve, where ACCURACY is measured as relative loss in accuracy to the original network, and PARAMETER-EFFICIENCY is measured by compression ratio. As there is no consensus on the definition of state-of-the-art in the pruning literature, we base our search on techniques listed in Ortiz et al. (2020).

**CIFAR-10 ResNet-56: Carreira-Perpiñán & Idelbayev (2018).** For the CIFAR-10 ResNet-56, we compare against "Learning Compression" (Carreira-Perpiñán & Idelbayev, 2018). This technique is selected as the most accurate technique at high sparsities, from Ortiz et al. (2020).

Carreira-Perpiñán & Idelbayev (2018) use unstructured gradual global magnitude pruning, derived from an alternating optimization formulation of the gradual pruning process which allows for weights to be reintroduced after being pruned. The pruning schedule is a hyperparameter, and the paper does not explain how the chosen value was found. After gradual pruning, Carreira-Perpiñán & Idelbayev fine-tune the remaining weights.

**ImageNet ResNet-50: He et al. (2018).** For the ImageNet ResNet-50, we compare against AMC (He et al., 2018). This technique is not listed in Ortiz et al. (2020), but achieves less reduction in accuracy at a higher sparsity than other techniques, losing $0.02\%$ top-1 accuracy at $5.13\times$ compression.

He et al. iteratively prune a ResNet-50, using manually selected per-iteration pruning rates (pruning by $50\%$, then $35\%$, then $25\%$, then $20\%$, resulting in a network that is $80.5\%$ sparse, a compression ratio of $5.13\times$), and retrain with fine-tuning for 30 epochs per iteration. On each pruning iteration, He et al. use a reinforcement-learning approach to determine layerwise pruning rates.

**WMT16 EN-DE GNMT: Zhu & Gupta (2018).** For the WMT16 EN-DE GNMT model, we compare against Zhu & Gupta (2018). This technique is not listed in Ortiz et al. (2020), as Ortiz et al. do not consider the GNMT model. To confirm that this technique is state-of-the-art on the GNMT model, we extensively searched among papers citing Zhu & Gupta (2018) and Wu et al. (2016) and found no results claiming a better ACCURACY versus PARAMETER-EFFICIENCY tradeoff curve.

Zhu & Gupta search across multiple pruning techniques, and ultimately use a pruning technique that prunes each layer at an equal rate, excluding the attention layers. Zhu & Gupta use a training

**ACCURACY versus PARAMETER-EFFICIENCY Tradeoff from Algorithm 1**

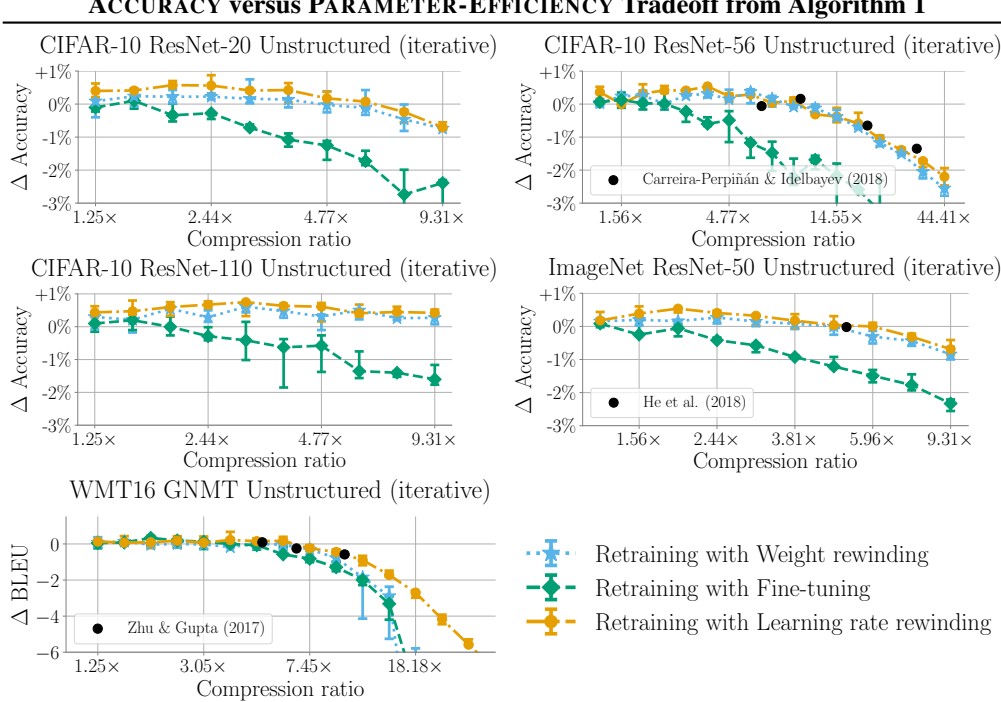

Figure 6: ACCURACY versus PARAMETER-EFFICIENCY tradeoff of our pruning algorithm.

algorithm that *gradually* prunes the network as it trains, using a specific polynomial to decide pruning rates over time, rather then fully training then pruning. Zhu & Gupta (2018) use a larger GNMT model than defined in the MLPerf benchmark, with 211M parameters to only 165M parameters in ours. Therefore, a model at a given compression ratio from Zhu & Gupta (2018) has more remaining parameters than a model at a given compression ratio using the GNMT model in this paper.

## D    OTHER INSTANTIATIONS OF OUR PRUNING ALGORITHMS

In this appendix, we present a comparison of the algorithm presented in Section 5 to instantiations of that algorithm with other retraining techniques. Specifically, we compare against retraining with weight rewinding for 90% of the original training time, learning rate rewinding for the original training time (as presented in the algorithm in the main body of the paper), or fine-tuning for the original training time.

---

**Algorithm 2** Our pruning algorithm

1. TRAIN to completion.
2. PRUNE the 20% lowest-magnitude weights globally.
3. RETRAIN using either weight rewinding for 90% of the original training time, learning rate rewinding for the original training time, or fine-tuning for the original training time.
4. Repeat steps 2 and 3 iteratively until the desired compression ratio is reached.

---

Figure 6 presents an evaluation of our pruning algorithm. We find that retraining with weight rewinding performs similarly well to retraining with learning rate rewinding, except for at high sparsities on the GNMT.

# E    ADDITIONAL NETWORKS AND BASELINES

In this appendix, we include results for more networks, pruning techniques, baselines, and ablations. We consider a larger set of networks than the paper, more structured pruning techniques from Li et al. (2017), a reinitialization baseline from Liu et al. (2019), and a natural ablation of rewinding, where we rewind the weights but use the learning rate of fine-tuning.

## METHODOLOGY

**Retraining techniques.** We include two baselines of the techniques presented in the main body of the paper, using the notation from Section 2. For convenience, we duplicate that notation here.

Neural network pruning is an algorithm that begins with a randomly initialized neural network with weights $W_0 \in \mathbb{R}^d$ and returns two objects: weights $W \in \mathbb{R}^d$ and a pruning mask $m \in \{0,1\}^d$ such that $W \odot m$ is the state of the pruned network (where $\odot$ is the element-wise product operator). Let $W_g$ be the weights of the network at epoch $g$. Let $T$ be the standard number of epochs for which the network is trained. Let $S[g]$ be the learning rate schedule of the network for each epoch $g$, defined such that $S[g > T] = S[T]$ (i.e., the last learning rate is extended indefinitely). Let $\text{TRAIN}^t(W, m, g)$ be a function that trains the weights of $W$ that are not pruned by mask $m$ for $t$ epochs according to the original learning rate schedule $S$, starting from step $g$.

LOW-LR WEIGHT REWINDING. The other natural ablation of weight rewinding (other than learning rate rewinding) is to rewind just the weights and use the learning rate that would have been used in fine-tuning. Formally, Low-LR weight rewinding for $t$ epochs runs $\text{TRAIN}^t(W_{T-t}, m, T)$.

REINITIALIZATION. We also consider reinitializing the discovered pruned network and retraining it by extending the original training schedule to the same total number of training epochs as fine-tuning trains for. Liu et al. (2019) found that for many pruning techniques, pruning and fine-tuning results in the same or worse performance as simply training the pruned network from scratch for an equivalent number of epochs. To address these concerns, we include comparisons against random reinitializations of networks with the discovered pruned structure, trained for the original $T$ training epochs plus the extra $t$ epochs that networks were retrained for. Formally reinitializing and retraining for $t$ epochs is sampling a new $W_0' \in \mathbb{R}^d$ then running $\text{TRAIN}^{T+t}(W_0', m, 0)$.

In this baseline, we consider the discovered structure from training and pruning the original network according to the given pruning technique. For unstructured pruning, this discovered structure is the specific structure left behind after magnitude pruning; for structured pruning, this discovered structure is the structure determined by the layerwise rates derived in Li et al. (2017). We note that in both of these cases, the resulting structure is determined by having trained the network, whether that occurs explicitly (as with unstructured pruning) or implicitly (as Li et al. determine layerwise pruning rates by pruning individual layers of a trained network). We therefore expect this to perform at least as well as randomly pruning the network before any amount of training, since the pruned structure incorporates knowledge from having already trained the network at least once.

**Networks, Datasets, and Hyperparameters.** We include two more CIFAR-10 vision networks in this appendix: ResNet-20 and ResNet-110. We also include several more structured pruning results from these networks, again given by Li et al. (2017): ResNet-56-{A,B} on CIFAR-10, ResNet-110-{A,B} on CIFAR-10, and ResNet-34-{A,B} on ImageNet. The networks and hyperparameters are described in Table 2.

**Data.** All plots are collected using the same methodology described in the main body of the paper. We also include the data from the networks presented in the main body of the paper for comparison. On each structured pruning plot, we plot the accuracy delta observed by Liu et al. (2019).

## RESULTS

**ACCURACY versus PARAMETER-EFFICIENCY tradeoff results (Figures 7 and 8).** Low-LR weight rewinding results in a large drop in accuracy relative to the best achievable accuracy, and a

---

[8] https://github.com/tensorflow/models/tree/v1.13.0/official/resnet

[9] https://github.com/tensorflow/tpu/tree/98497e0b/models/official/resnet

[10] https://github.com/mlperf/training_results_v0.5/tree/7238ee7/v0.5.0/google/cloud_v3.8/gnmt-tpuv3-8

| Dataset | Network | #Params | Optimizer | Learning rate (t = training epoch) | | Accuracy |
|---------|---------|---------|-----------|-----------------------------------|--|----------|
| CIFAR-10 | ResNet-20[8] | 271K | Nesterov SGD $\beta = 0.9$ Batch size: 128 Weight decay: 0.0002 Epochs: 182 | $\alpha = \begin{cases} 0.1 & t \in [0, 91) \\ 0.01 & t \in [91, 136) \\ 0.001 & t \in [136, 182] \end{cases}$ | | $91.71 \pm 0.23\%$ |
| | ResNet-56[8] | 852K | | | | $93.46 \pm 0.21\%$ |
| | ResNet-110[8] | 1.72M | | | | $93.77 \pm 0.23\%$ |
| ImageNet | ResNet-34[9] | 21.8M | Nesterov SGD $\beta = 0.9$ Batch size: 1024 Weight decay: 0.0001 Epochs: 90 | $\alpha = \begin{cases} 0.4 \cdot \frac{1}{5} & t \in [0, 5) \\ 0.4 & t \in [5, 30) \\ 0.04 & t \in [30, 60) \\ 0.004 & t \in [60, 80) \\ 0.0004 & t \in [80, 90] \end{cases}$ | | $73.60 \pm 0.27\%$ top-1 |
| | ResNet-50[9] | 25.5M | | | | $76.17 \pm 0.14\%$ top-1 |
| WMT16 EN-DE | GNMT[10] | 165M | Adam $\beta_1 = 0.9$ $\beta_2 = 0.999$ Batch size: 2048 Epochs: 5 | $\alpha = \begin{cases} 0.002 \cdot 0.01^{1-8t} & t \in [0, 0.125) \\ 0.002 & t \in [0.125, 3.75) \\ 0.001 & t \in [3.75, 4.165) \\ 0.0005 & t \in [4.165, 4.58) \\ 0.00025 & t \in [4.58, 5) \end{cases}$ | | `newstest2015:` $26.87 \pm 0.23$ BLEU |

Table 2: Networks, datasets, and hyperparameters. We use standard implementations available online and standard hyperparameters. All accuracies are in line with baselines reported for these networks (Liu et al., 2019; He et al., 2018; Gale et al., 2019; Wu et al., 2016; Zhu & Gupta, 2018).

small drop in accuracy compared to standard fine-tuning. With unstructured pruning, reinitialization performs poorly relative to all other retraining techniques. With structured pruning, reinitialization performs much better, roughly matching the performance of rewinding the weights and learning rate. This is expected from the results of Liu et al. (2019), which find that reinitialization comparatively performs well with structured pruning techniques.

**ACCURACY versus SEARCH COST tradeoff results (Figures 9 and 10).** Low-LR weight rewinding has markedly different behavior than other techniques when picking where to rewind to. Specifically, when performing low-LR weight rewinding, longer training does not always result in higher accuracy. Instead, accuracy peaks at different points on different networks and compression ratios, often when rewinding to near the middle of training.

Reinitialization typically saturates in accuracy with the original training schedule, and does not gain a significant boost in accuracy from adding extra retraining epochs. When performing structured pruning, this means that reinitialization achieves the highest accuracy with few retraining epochs, although rewinding the learning rate can still achieve higher accuracy than reinitialization with sufficient training.

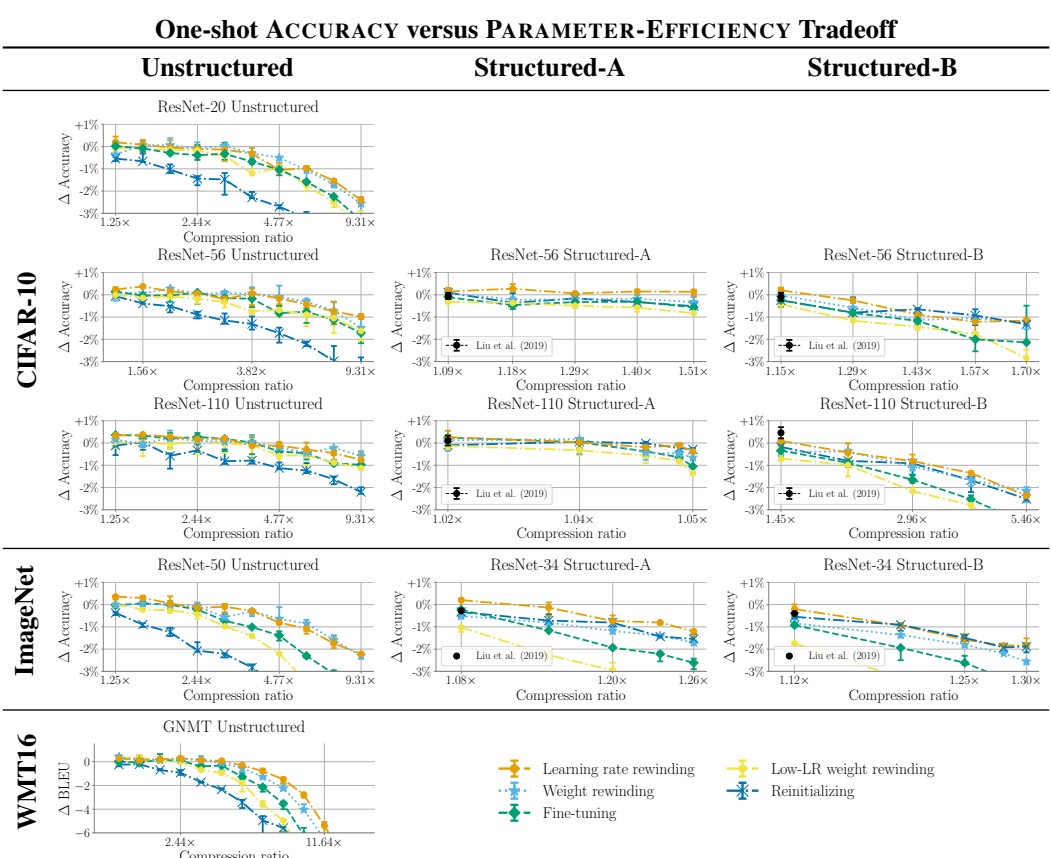

Figure 7: The best achievable accuracy across retraining times by one-shot pruning.

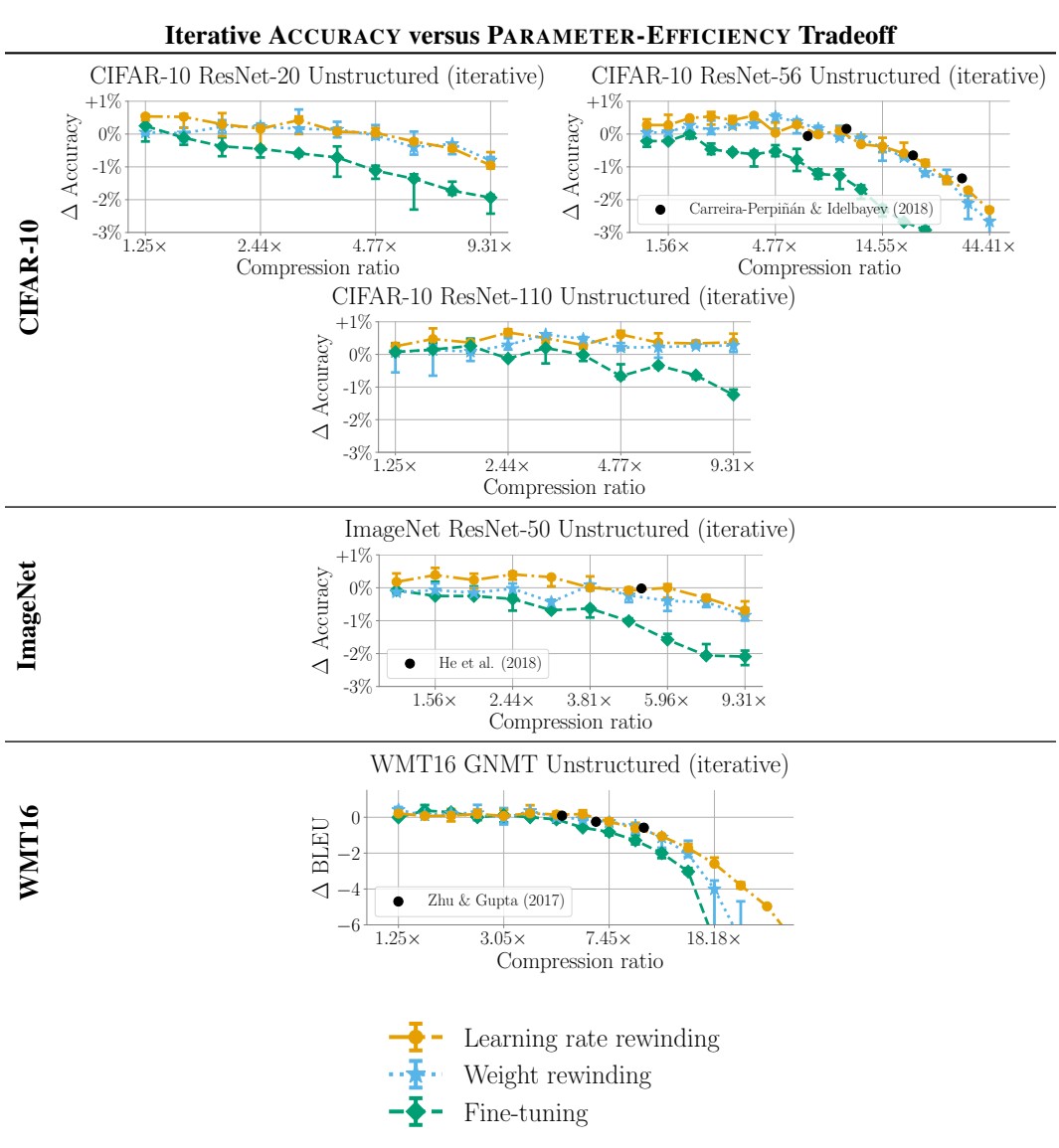

Figure 8: The best achievable accuracy across retraining times by iteratively pruning.

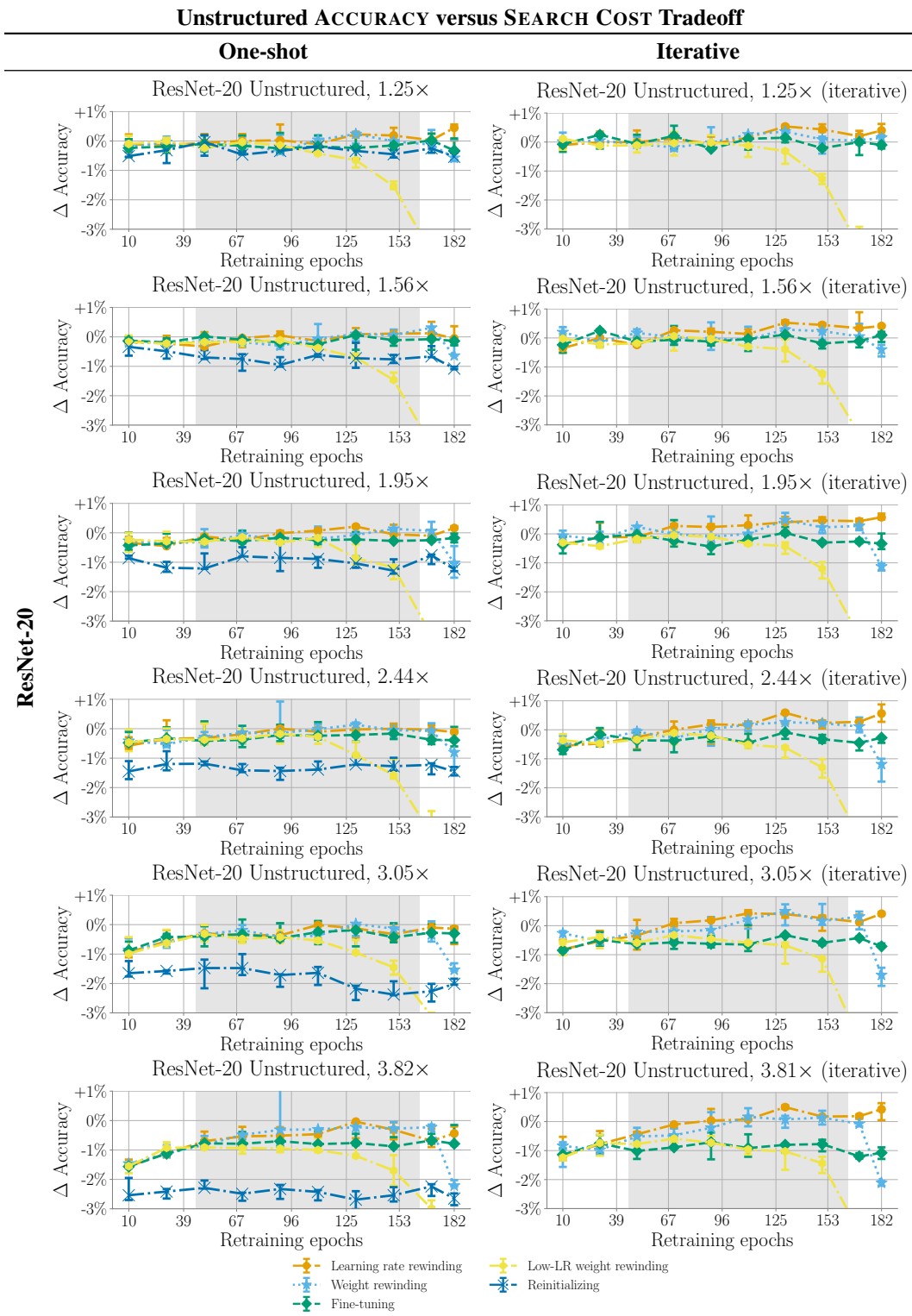

Figure 9: Accuracy curves across different networks and compressions using unstructured pruning.

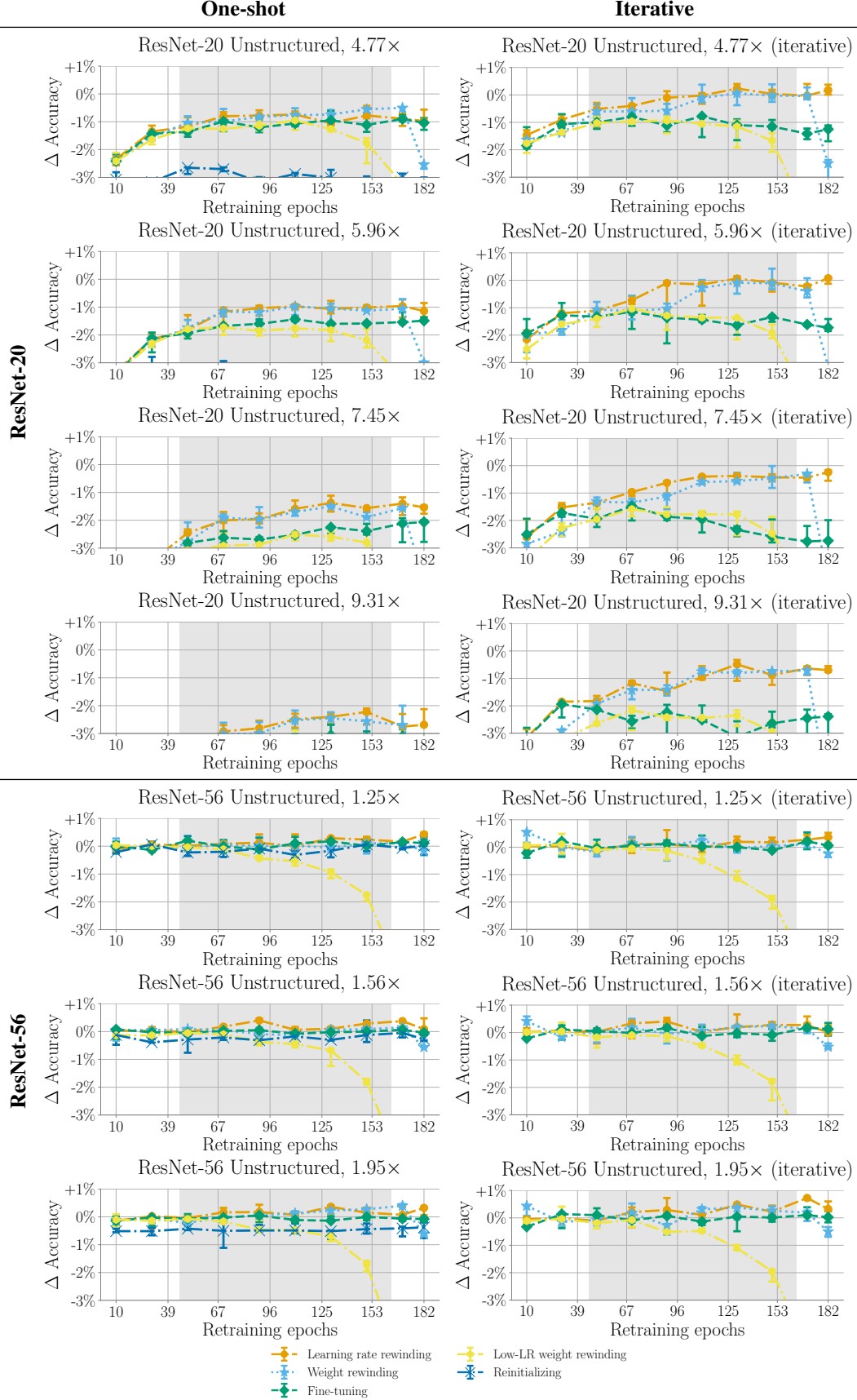

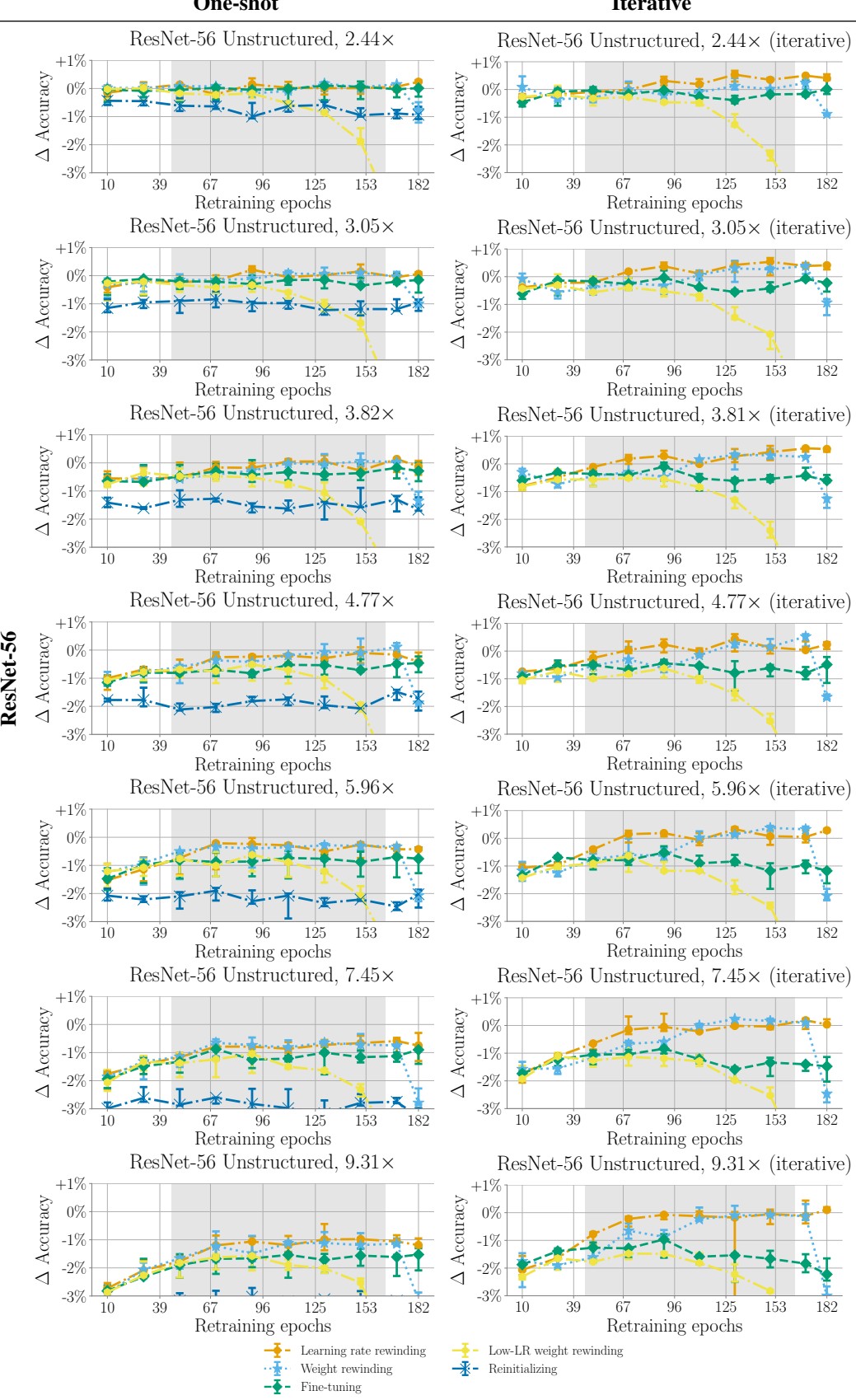

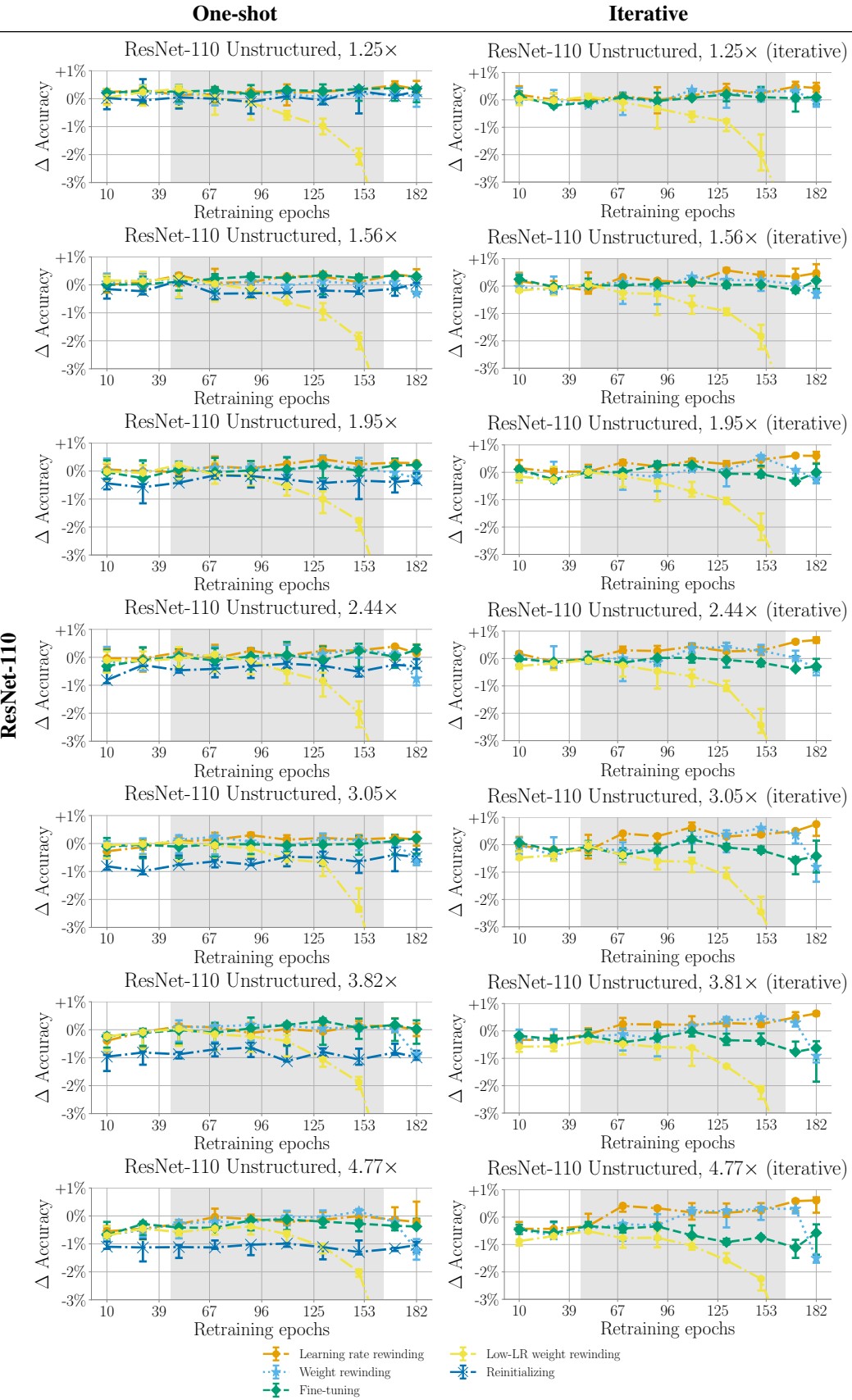

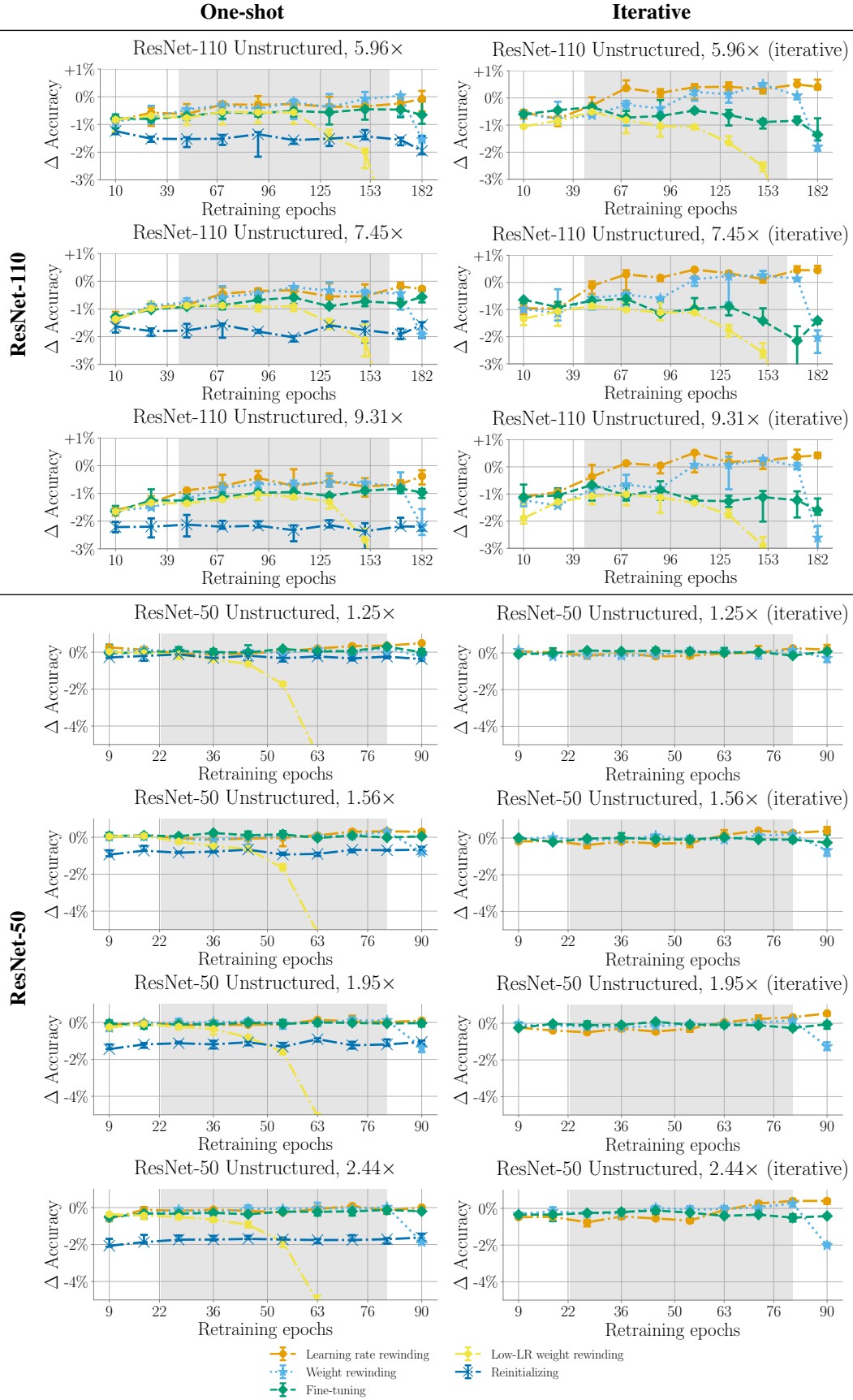

**One-shot**  **Iterative**

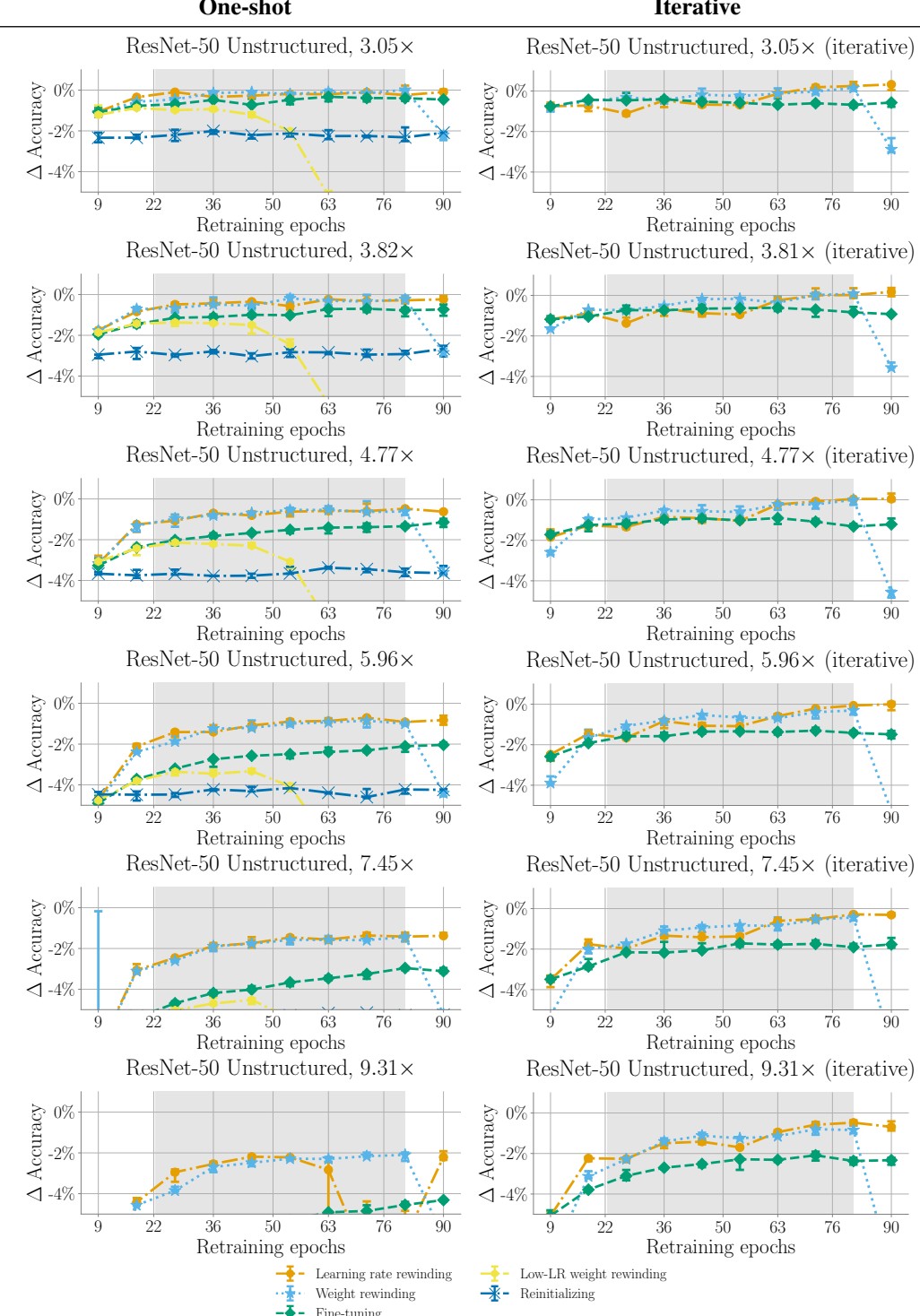

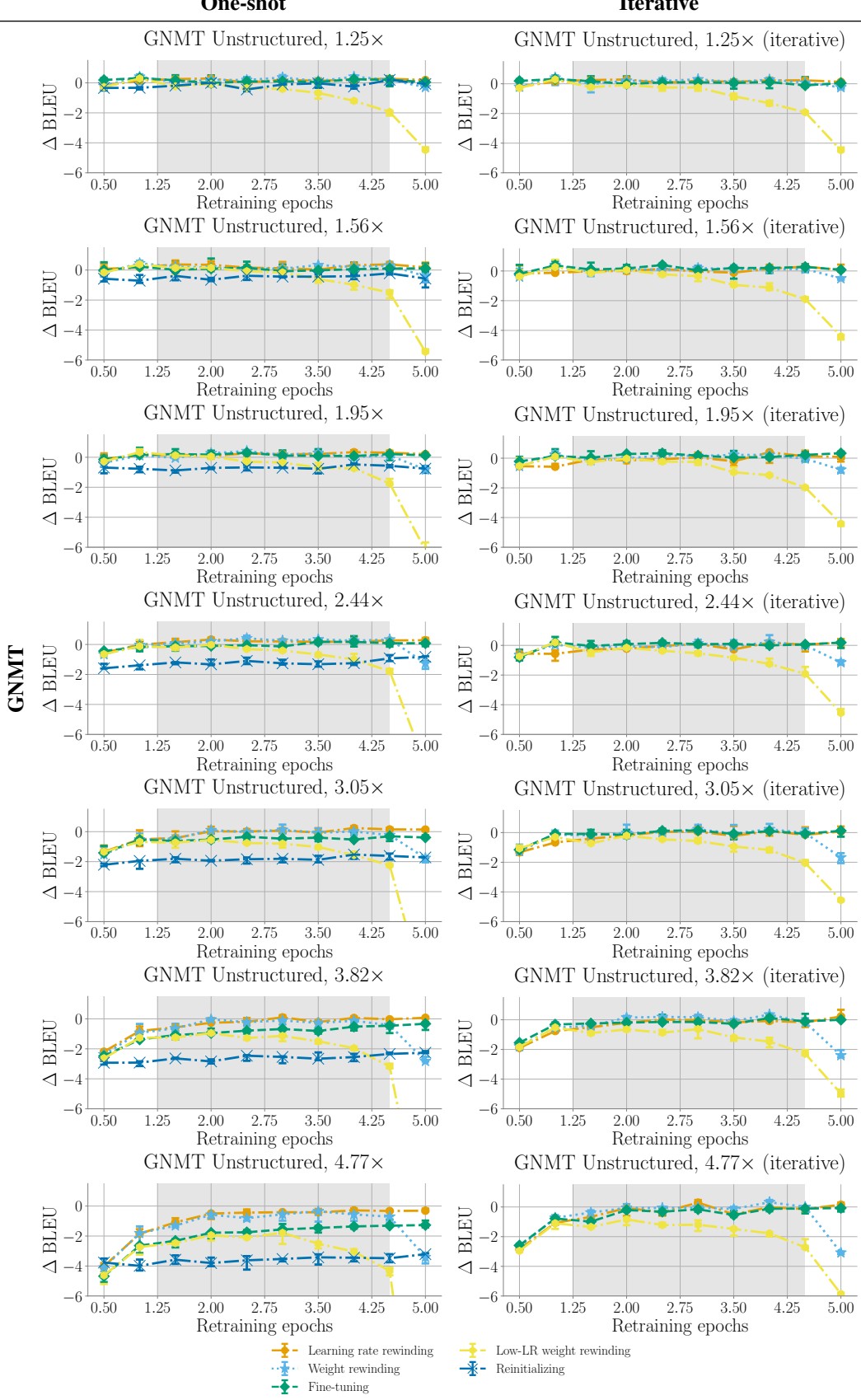

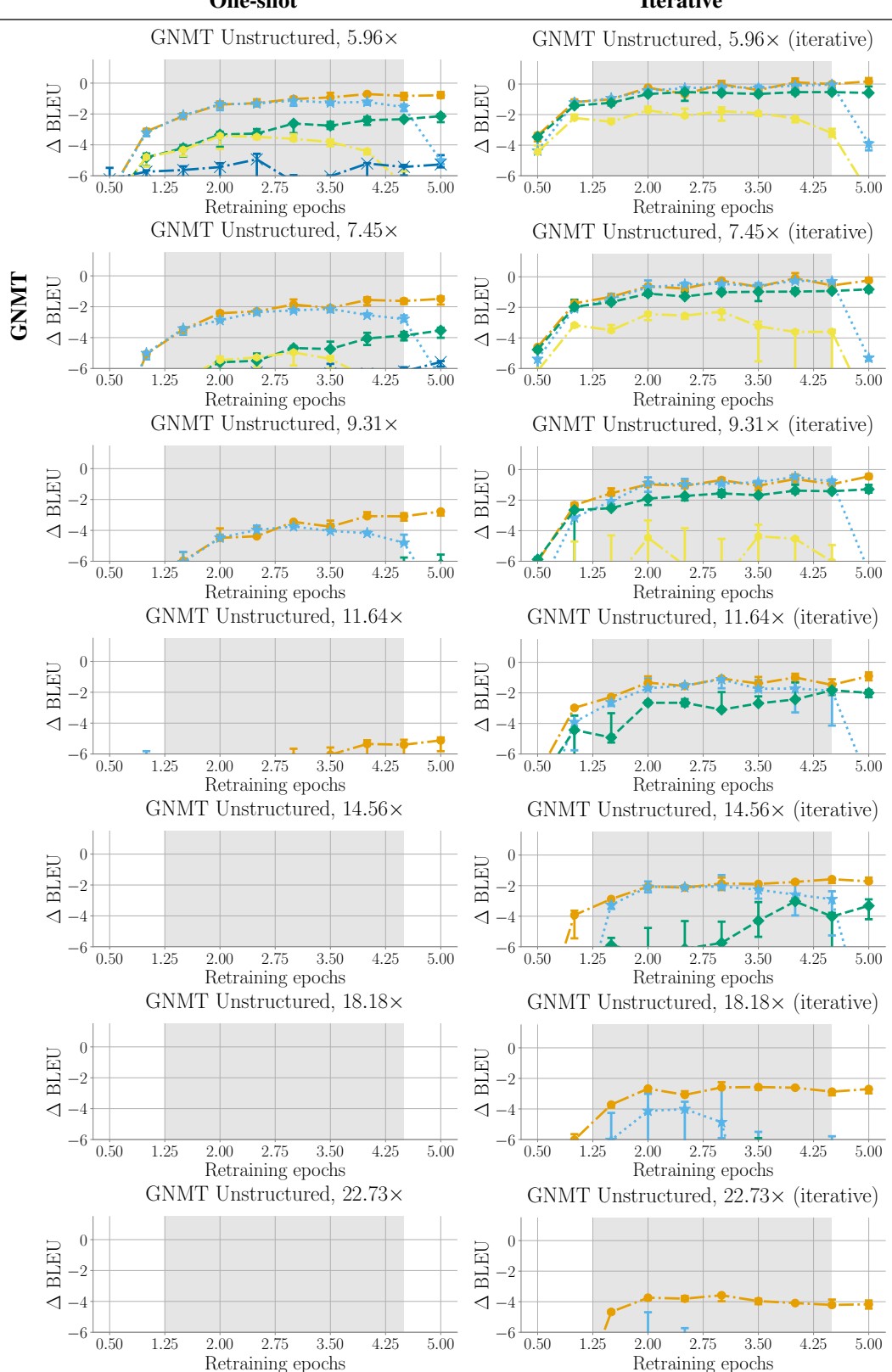

**Structured ACCURACY versus SEARCH COST Tradeoff**

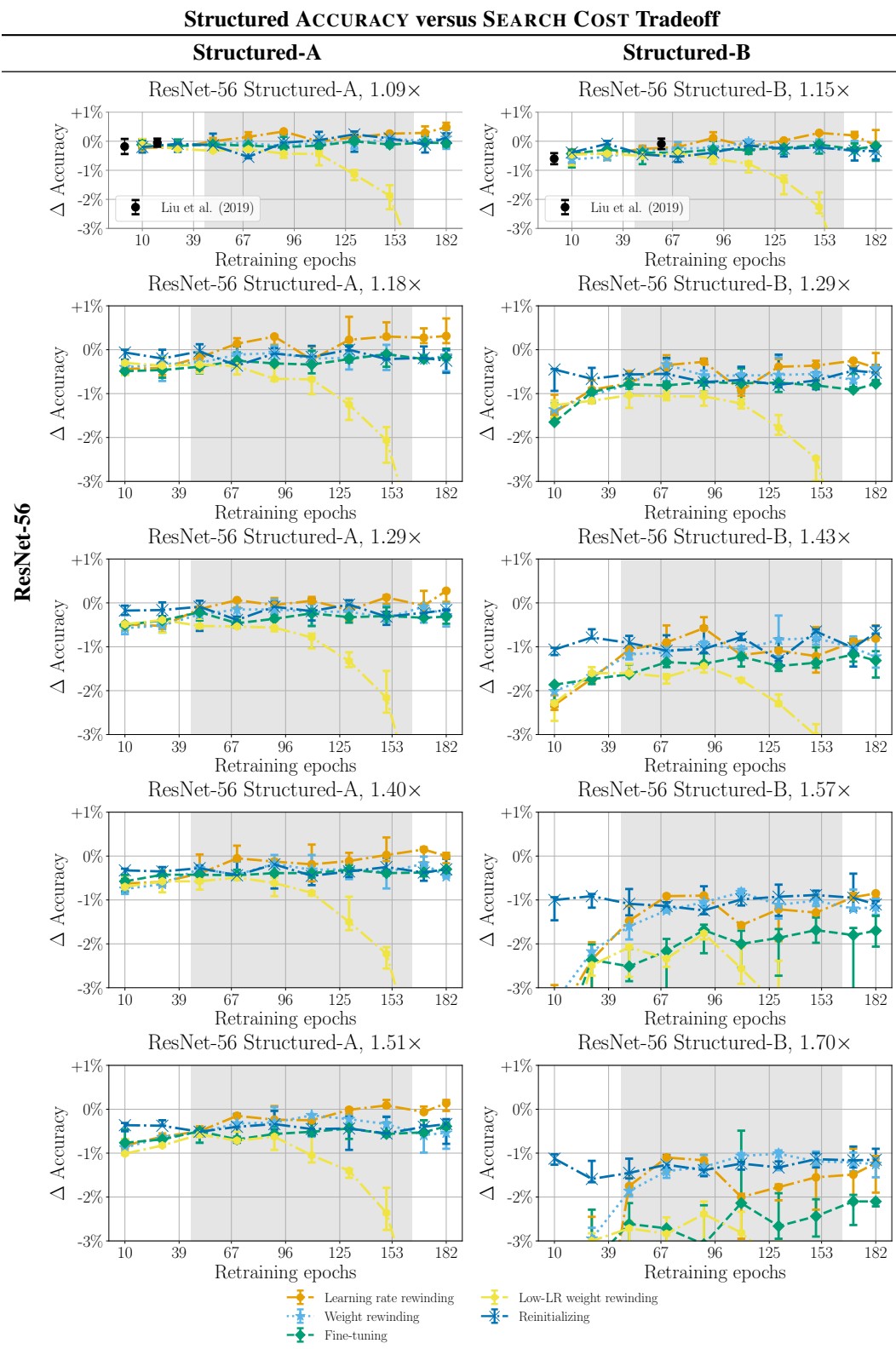

Figure 10: Accuracy curves across different networks and compressions using structured pruning.

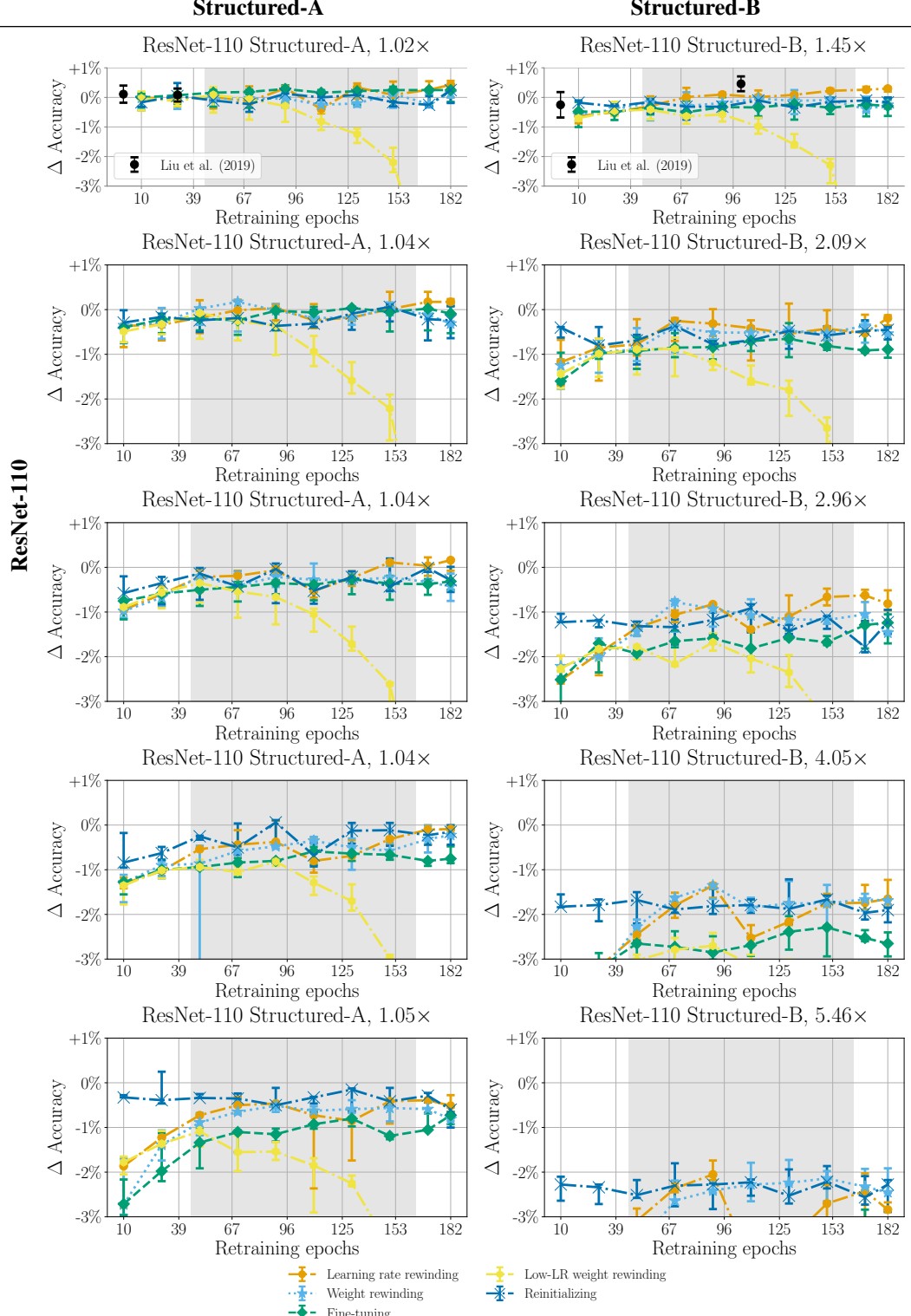

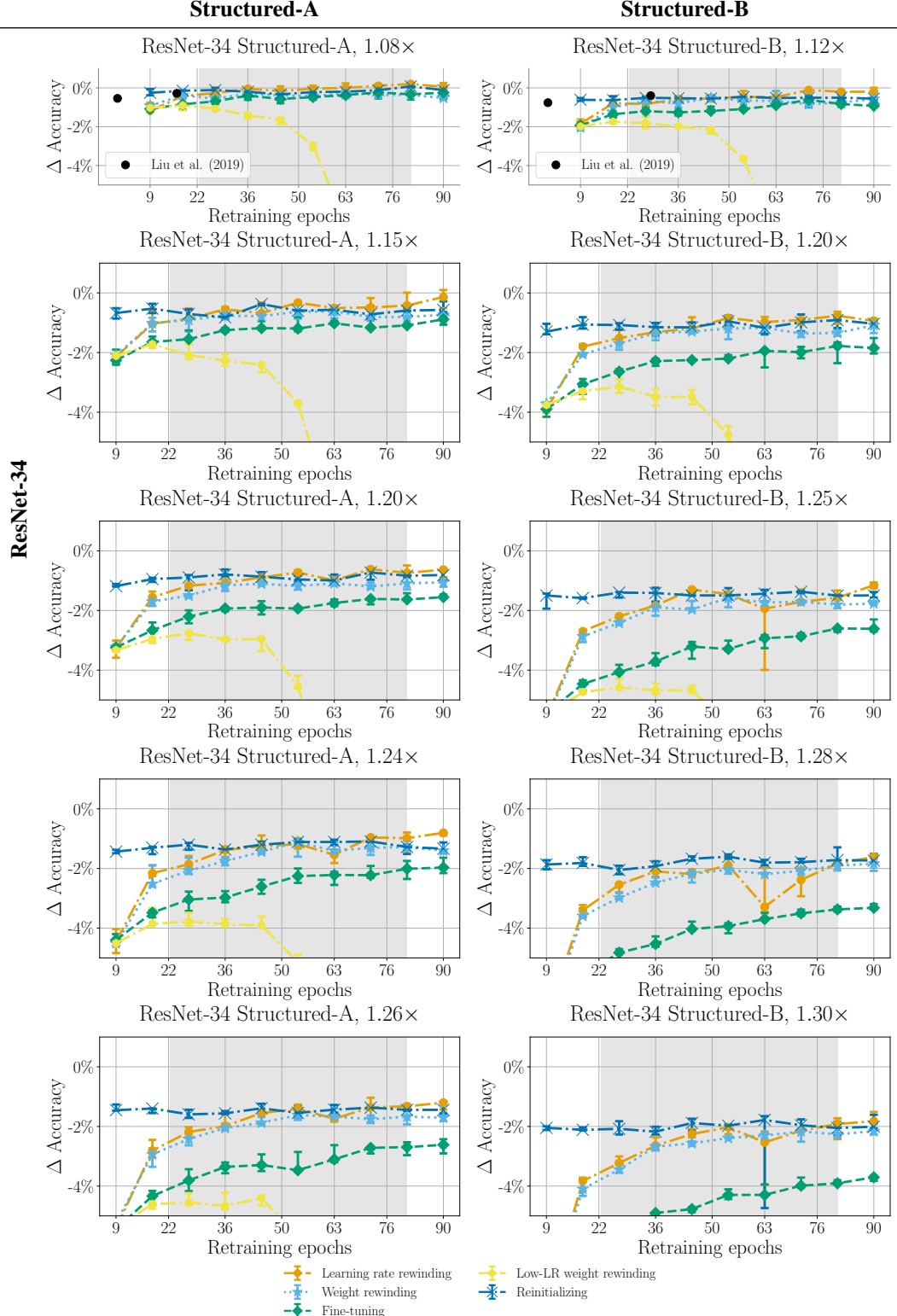

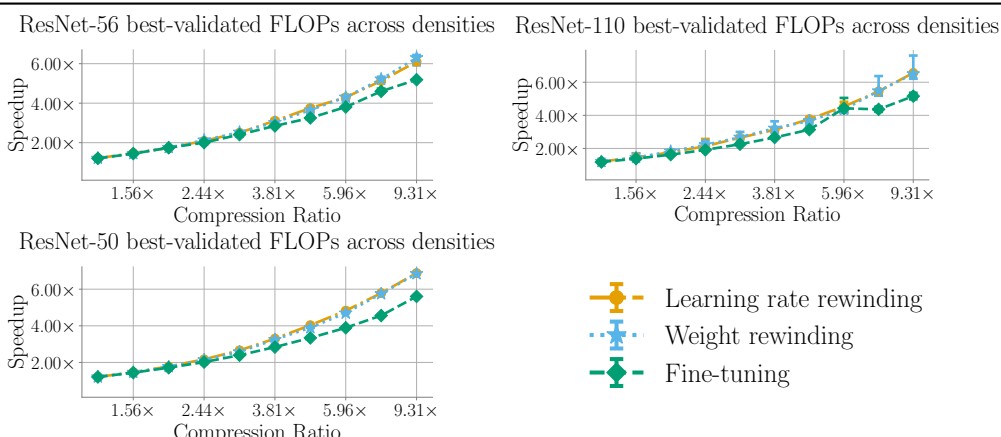

Figure 11: Speedup over original network for different retraining techniques and networks

**Iterative pruning FLOP speedup over fine-tuning**

Figure 12: Speedup over original network for different retraining techniques and networks

## F  COMPRESSION RATIO VS FLOPS

In the main body of the paper, we use the compression ratio as the metric denoting the EFFICIENCY of a given neural network. However, the compression ratio does not tell the full story: networks of different compression ratios can require different amounts of floating point operations (FLOPs) to perform inference. For instance, pruning a weight in the first convolutional layer of a ResNet results in pruning more FLOPs than pruning a weight in the last convolutional layer. Reduction in FLOPs can (but does not necessarily) result in wall clock speedup (Baghdadi et al., 2019). In this appendix, we analyze the number of FLOPs required to perform inference on pruned networks acquired through fine-tuning and rewinding. Our methodology for one-shot pruning uses the same initial trained network for our comparisons between all techniques, and prunes using the same pruning technique. This means that both networks have the exact same sparsity pattern, and therefore same number of FLOPs. For iterative pruning the networks diverge, meaning the FLOPs also differ, since we use global pruning.

**Methodology.** In this appendix, we compare the FLOPs between different retraining techniques. We specifically consider iteratively pruned vision networks, where pruning weights in earlier layers results in a larger reduction in FLOPs than pruning weights in later layers. We compare using the

same networks as selected in the iterative subsection of Section 3, i.e. the using the retraining time that results in the set of networks with the highest validation accuracy for each different compression ratio. Therefore the FLOPs reported in this section are the FLOPs resulting from the most accurate network at a given compression ratio, not necessarily the minimum required FLOPs at that compression ratio. In Figure 11, we plot the theoretical speedup over the original network – i.e., the ratio of original FLOPs over pruned FLOPs. In Figure 12, we plot the theoretical speedup of each technique over fine-tuning – i.e., the ratio of fine-tuning FLOPs at that compression ratio to the FLOPs of each other technique at that compression ratio.

**Results.** Both rewinding techniques find networks that require fewer FLOPs than those found by iterative pruning with standard fine-tuning. Due to the increased accuracy from rewinding, this results in a magnified decrease in INFERENCE-EFFICIENCY for rewinding compared to fine-tuning. For instance, a ResNet-50 pruned to maintain the same accuracy as the original network results in a $4.8\times$ theoretical speedup from the original network with rewinding the learning rate, whereas a similarly accurate network attained through fine-tuning has a speedup of $1.7\times$.

