# OpenReview forum: "Comparing Rewinding and Fine-tuning in Neural Network Pruning"
_ICLR.cc/2020/Conference — Accept (Talk)_

### Official Review · AnonReviewer1 · 2019-10-21
**Official Blind Review #1**

**Rating:** 8

**Review:**

This paper does an in-depth evaluation of the notion of rewinding pruned networks to the weights at a previous point in training, and then re-training the pruned network from then on. This is in comparison with fine-tuning a pruned network, where the retraining continues from the network's current weights. The authors focused on vision networks and unstructured magnitude pruning in their evaluation.

The paper is well-written and easy to follow, and the authors have done a good job of empirically comparing rewinding against fine-tuning in a number of different scenarios.

My main concern with the paper is that it seems too incremental to stand on its own. Rewinding is a notion that was already explored in the Lotter Ticket Hypothesis (Frankle et al., 2019), so this paper seems to be more of an extension of that work. The take-away message from this paper (stated by authors in their conclusion) is that practicioners "should explore rewinding as an alternative to fine-tuning for neural network pruning", but that's a case that was already made by the LTH paper. The current work certainly gives more weight to that claim, but I don't feel the contributions are strong enough on their own to justify a full conference publication.

I encourage the authors to continue working on this, as it is very interesting and can be very useful. Some ideas to make the paper stronger:
- Given that this is a purely empirical paper, it'd be better to not limit the experiments as much. Can you run on non-vision networks? What about mobile-net? Can you try different pruning techniques? etc.
- How do the different methods compare in terms of accuracy/sparsity versus FLOPs?

Finally, two minor comments to improve the writing:
- First sentence of 3.1: s/that meet that the accuracy/that match the accuracy/
- First sentence of 3.1: s/than fine-tuning can/compared to fine tuning

**Experience Assessment:**

I have read many papers in this area.

**Review Assessment: Checking Correctness Of Derivations And Theory:**

N/A

**Review Assessment: Checking Correctness Of Experiments:**

I assessed the sensibility of the experiments.

**Review Assessment: Thoroughness In Paper Reading:**

I read the paper thoroughly.

---

> ### Author Response · Authors · 2019-11-15
> **Response to Reviewer 1**
>
> > My main concern with the paper is that it seems too incremental to stand on its own...
>
> Frankle et al. do not make the case that practitioners should explore rewinding as an alternative to fine-tuning, as they do not provide any comparisons against fine-tuning. Please see our overall response above for a full clarification of the novelty of our work, and its relationship to others' work.
>
> > Given that this is a purely empirical paper, it'd be better to not limit the experiments as much...
>
> Please see the overall response above for new sets of experiments, including machine translation and structured pruning.
>
> > How do the different methods compare in terms of accuracy/sparsity versus FLOPs?
>
> This is a great question, and drew out an interesting set of new results, which are included in the overall response above.
>
> > Finally, two minor comments to improve the writing...
>
> Thanks for the feedback! We have incorporated the changes.

---

### Official Review · AnonReviewer2 · 2019-10-24
**Official Blind Review #2**

**Rating:** 6

**Review:**

Building on the results of [Frankle et al, 2019], this paper seeks to utilize rewinding as a core procedure in pruning neural networks, in combination with the usual fine-tuning procedures.  Specifically, [Frankle et al, 2019] demonstrate that it is possible to find sparse subnetworks, such that rewinding weights to their initial values and retraining from that initialization, yields test accuracy similar to the original network.  While this is already a form of pruning, the submitted paper explores a wider space of pruning procedures that utilize rewinding as a subroutine.

This wider framework includes the choice of rewind point (e.g. rewinding to partway through training rather than to initialization) and how to balance computation budget between rewinding (and retraining for an equivalent number of epochs) vs continuing to fine-tune a network.  Experiments cover this hyperparameter space, as well as the range of desired sparsity level (pruning amount).  Results show rewinding (to a point 30% - 60% into training) dominates any amount of fine-tuning, if moderate to high sparsity is desired.

The empirical study conducted by this paper is useful and complements the results previously reported in [Frankle et al, 2019].  However, the paper itself is light on novelty, as the core ideas were already established by [Frankle et al], and the application of them here is relatively straightforward.  The extensive experiments here add value to the conversation about the lottery ticket hypothesis, but are not otherwise ground-breaking.


**Experience Assessment:**

I have read many papers in this area.

**Review Assessment: Checking Correctness Of Derivations And Theory:**

N/A

**Review Assessment: Checking Correctness Of Experiments:**

I carefully checked the experiments.

**Review Assessment: Thoroughness In Paper Reading:**

I read the paper thoroughly.

---

> ### Author Response · Authors · 2019-11-15
> **Response to Reviewer 2**
>
> > However, the paper itself is light on novelty...
>
> Please see our overall response above for a full clarification of the novelty of our work, and its relationship to others' work.

---

### Official Review · AnonReviewer3 · 2019-11-04
**Official Blind Review #3**

**Rating:** 8

**Review:**

*Summary*
Extending the observations of Frankle et al. (2019, "The Lottery Ticket Hypothesis"), this paper examines "rewinding" as an alternative to fine-tuning in a typical network pruning process. After training to convergence for T iterations, the k% of weights with the smallest magnitude are pruned (set to zero). Typically, in fine-tuning, the remaining weights are continued to be trained for several more iterations at a small learning rate. With "rewinding", the remaining weights are reset to their values in iteration 0<=t<T and are trained for T-t iterations with the original learning rate schedule.

Experiments with this method (and an iterative variant) show that (a) rewinding achieves 2-5x smaller networks matching the unpruned accuracy (with comprehensive tuning of the rewinding parameter t); (b) selection of rewinding iteration is important, but is flexible within a large range for higher sparsity pruning.

*Rating*
The paper is very well written with good exposition, thorough notes and citations for all methodological choices,
and an explicit statement of the limitations of the work.

A few considerations relevant to the rating:
(1) Novelty: The idea for rewinding is not novel, as acknowledged clearly in the paper. Frankle et al. (2019, "Stabilizing the Lottery Ticket Hypothesis") showed that rewinding to an early iteration of training (0<t<T) yielded better accuracy than rewinding to iteration t=0 for VGG-19 and ResNet-18 on CIFAR-10. However, Frankle et al. did not consider fine-tuning as it was not relevant to lottery ticket discovery. This submission studies the tradeoffs of rewinding vs. fine-tuning.

(2) Thoroughness: The paper considers ResNet-20 and VGG-16 with CIFAR-10 and ResNet-50 with ImageNet. Conclusions would be strengthened with additional combinations of networks and datasets.

(3) Acknowledged limitations: As noted, the paper doesn't consider any pruning criteria other than weight magnitude, nor does it consider structured pruning. The latter in particular is important for applications where prediction speed on commodity hardware is a limiting factor.

As it is, I think this paper a worthy (if limited) contribution to the understanding of network pruning.

*Notes*
Table 1/Figures *: note which dataset is used for each architecture
Figures 2-3: It seems that many values are clipped by the legend range of +/- 0.5%. Consider showing the figure with a larger range or adding such a figure to the appendix.

**Experience Assessment:**

I have published one or two papers in this area.

**Review Assessment: Checking Correctness Of Derivations And Theory:**

N/A

**Review Assessment: Checking Correctness Of Experiments:**

I assessed the sensibility of the experiments.

**Review Assessment: Thoroughness In Paper Reading:**

I read the paper at least twice and used my best judgement in assessing the paper.

---

> ### Author Response · Authors · 2019-11-15
> **Response to Reviewer 3**
>
> > (1) Novelty...
>
> Please see our overall response above for a full clarification of the novelty of our work, and its relationship to others' work.
>
> > (2) Thoroughness... (3) Acknowledged limitations...
>
> Please see the overall response above for new sets of experiments, including machine translation and structured pruning.
>
> > Notes...
>
> Thanks for the notes! We will incorporate the changes in the final version of the paper.

---

### Author Response · Authors · 2019-11-15
**Response to all reviewers**

# Novelty

We believe that we didn't fully articulate the novelty of our results in the submission.

While this paper does not present a novel algorithm, it does presents novel insight, experiments, and analysis of the behavior of an existing algorithm, all of which are outside of the scope of the algorithm's original analysis. When Frankle et al. [1] introduced rewinding, they did not evaluate the efficacy of the algorithm as a pruning technique. Instead, they use it as a technique to analyze sparse neural network stability (robustness to noise during training), rewinding to at most 30% of the way through training in their experiments. Our analysis instead rewinds all throughout training, and then compares the resulting networks to those of fine-tuning for an equivalent number of epochs. Our analysis therefore does not overlap with, and does not follow from, the analysis in their paper.

Our results demonstrate that rewinding is an effective pruning technique, and in certain settings can be a higher performing drop-in replacement for fine-tuning, a claim that Frankle et al. do not propose or evaluate. Moreover, these results are not to be expected, given that rewinding goes backwards in the training processes, whereas fine-tuning adds on additional training epochs.


# Structured pruning (Reviewers 1 and 3)

We implemented Li et al. [2]'s filter pruning technique, and compared fine-tuning and rewinding using per-layer pruning rates reported in their paper. To compare across sparsity levels, we also apply the pruning rates iteratively to obtain sparser networks. We use the rates in Table 1 in [2], specifically VGG-16-pruned-A (CIFAR), ResNet-56-pruned-{A,B} (CIFAR), ResNet-110-pruned-{A,B} (CIFAR), and ResNet-34-pruned-{A,B} (ImageNet). The rest of our hyperparameters remain identical to our original experiments.

Appendix B of the updated draft of the paper presents our results. We find that rewinding outperforms fine-tuning in the structured pruning setting. At the exact sparsity levels reported in [2], the techniques are indistinguishable. However, for sparser networks, rewinding outperforms fine-tuning, both in terms of epoch-for-epoch accuracy and max achievable accuracy. These results are consistent with our findings for unstructured pruning.

# Non-vision network (Reviewers 1 and 3)

We compared fine-tuning and rewinding on the GNMT model (https://arxiv.org/abs/1609.08144) in the MLPerf benchmark (https://mlperf.org/training-overview), which is a seq-to-seq model based on stacked LSTMs, trained on the WMT English-German dataset. We used NVIDIA's implementation at https://github.com/NVIDIA/DeepLearningExamples/tree/4e00153/TensorFlow/Translation/GNMT. Due to hardware and time constraints, we use hidden layer sizes of 512 units rather than 1024 units. We prune all weights using global magnitude pruning. We compare across a small set of sparsities and re-training budgets.

Our results are consistent with the results for vision networks, showing that at high sparsities (80%), rewinding outperforms fine-tuning on this LSTM-based network. At lower sparsities, fine-tuning outperforms rewinding, and at medium sparsities, the two techniques behave approximately equivalently. These results match those seen for vision networks, and are visualized in Appendix C.

# FLOP comparison (Reviewer 1)

Reviewer 1:
> How do the different methods compare in terms of accuracy/sparsity versus FLOPs?

This is a good question. We perform this comparison and include data and discussion in Appendix D of the updated draft of the paper.


In sum, for one-shot pruning, we use the same initial trained network for our comparisons between both methods, and prune using the same pruning technique. This means that both networks have the exact same sparsity pattern, and therefore same number of FLOPs. For iterative pruning the networks differ, meaning the FLOPs also differ, since we use a global pruning technique. Our results show that at any given sparsity, rewinding results in lower FLOP networks than fine-tuning: the techniques are roughly equivalent at low sparsities, but rewinding results in up to a 1.25x speedup at high sparsities. We will include FLOP counts for all networks in the final version of the paper.

[1] Frankle et al. Stabilizing the Lottery Ticket Hypothesis. Arxiv.
[2] Li et al. Pruning Filters for Efficient ConvNets. ICLR, 2017.

---

### Author Response · Authors · 2020-03-03
**Uploaded camera-ready version**

We have made the following changes to improve the paper and address concerns raised in the reviews:
# Changes requested by reviewers
- (Reviewers 1 and 3): We have included more networks, datasets, and pruning methods. In particular, we include structured pruning (from Li et al. [1]) and a GNMT model (from Wu et al. [2]) trained on WMT16 EN-DE.
- (Reviewer 1): We have included an analysis of the comparison of the methods based on their resultant FLOP-counts in Appendix F.
These results are consistent with the results initially reported in the rebuttal.


# Additional results
- Since submitting the paper, we have found that an alternative version of weight rewinding, which we name learning rate rewinding (rewinding just the learning rate and not the weights), outperforms weight rewinding by a small amount. We have updated our paper to additionally include this technique. The findings in the original submission still hold, and are still presented in the updated draft -- that weight rewinding outperforms standard fine-tuning, can match state-of-the-art results achieved by more complex techniques, and performs well for a wide range of hyperparameter choices.


[1] Li et al. Pruning Filters for Efficient ConvNets. ICLR, 2017.
[2] Wu et al. Google's Neural Machine Translation System: Bridging the Gap between Human and Machine Translation. arXiv:1609.08144, 2016.

---

### Decision · Program_Chairs · 2019-12-19

**Decision:**

Accept (Talk)

**Comment:**

Reviewers unanimously accepted this paper.